# TET3 prevents terminal differentiation of adult NSCs by a non-catalytic action at *Snrpn*

Raquel Montalbán-Loro[1], Anna Lozano-Ureña[1], Mitsuteru Ito [2], Christel Krueger[3], Wolf Reik[3,4], Anne C. Ferguson-Smith[2] & Sacri R. Ferrón [1]

Ten-eleven-translocation (TET) proteins catalyze DNA hydroxylation, playing an important role in demethylation of DNA in mammals. Remarkably, although hydroxymethylation levels are high in the mouse brain, the potential role of TET proteins in adult neurogenesis is unknown. We show here that a non-catalytic action of TET3 is essentially required for the maintenance of the neural stem cell (NSC) pool in the adult subventricular zone (SVZ) niche by preventing premature differentiation of NSCs into non-neurogenic astrocytes. This occurs through direct binding of TET3 to the paternal transcribed allele of the imprinted gene Small nuclear ribonucleoprotein-associated polypeptide N (*Snrpn*), contributing to transcriptional repression of the gene. The study also identifies BMP2 as an effector of the astrocytic terminal differentiation mediated by SNRPN. Our work describes a novel mechanism of control of an imprinted gene in the regulation of adult neurogenesis through an unconventional role of TET3.

[1] ERI BiotecMed/Departamento de Biología Celular, Universidad de Valencia, 46100 Valencia, Spain. [2] Department of Genetics, University of Cambridge, Cambridge CB2 3EH, UK. [3] Epigenetics Programme, The Babraham Institute, Cambridge CB22 3AT, UK. [4] Wellcome Trust Sanger Institute, Cambridge CB10 1SA, UK. These authors contributed equally: Raquel Montalbán-Loro and Anna Lozano-Ureña. Correspondence and requests for materials should be addressed to S.R.F. (email: sacramento.rodriguez@uv.es)

In the mammalian brain two regions generate new neurons throughout adulthood: the subventricular zone (SVZ) in the walls of the lateral ventricles and the subgranular zone (SGZ) of the dentate gyrus (DG) in the hippocampus[1,2]. The process of neurogenesis in these adult neurogenic niches is continually sustained by the activity of neural stem cells (NSCs) which are characterized by their ability to balance self-renewal with multi-potential differentiation into astrocytes, oligodendrocytes, and neurons[3]. NSCs are lineage-related to radial glial cells, therefore, they exhibit astrocytic characteristics, such as glial fibrillary acidic protein (GFAP)+filaments or astrocyte-specific glutamate aspartate transporters (GLAST)[4]. NSCs also express the transcription factor *Sox2* (Sry-related HMG box 2)[4,5]. In the SVZ in particular, NSCs also known as type B cells have a long basal process contacting blood vessels[6] and extend an apical process ending in a primary cilium that protrudes into the ventricle[7]. The walls of the lateral ventricles thus show a typical organization where the apical process of type B cells are surrounded by a rosette of epithelial ependymal cells forming structures known as pinwheels[8]. Activated and quiescent NSCs appear to coexist in the SVZ[9]. Once activated, NSCs give rise to transit-amplifying progenitors or type C-cells, which in turn, generate neuroblasts, or type A-cells. These cells migrate anteriorly forming the rostral migratory stream (RMS) to the olfactory bulb (OB), where they mature into functional neurons involved in olfactory discrimination[10]. Subventricular NSCs are also capable of producing some oligodendroblasts that migrate to the *corpus callosum*[11,12].

Signaling between NSCs and other cells of their microenvironment or niche is critical for adult neurogenesis, through its interaction with NSCs intrinsic programs of gene expression[9]. Spatial and temporal gene regulation of gene expression in NSCs is established and maintained by the coordinated interaction between transcription factors and epigenetic regulators. Most mammalian genes are expressed from both maternally and paternally inherited chromosomal homologs. In contrast, imprinted genes are expressed from one parental copy only and the gene copy inherited from the other progenitor remains repressed[13]. Their monoallelic expression makes these *loci* very vulnerable as mutation or deregulation of the sole expressed allele can compromise expression and lead to severe developmental defects[14,15]. Interestingly, we have previously shown that selective absence of imprinting can occur in particular lineages to modulate the dosage of imprinted genes for cell-specific functions in physiological contexts[16,17]. These context-dependent changes may be important for cell plasticity during normal development and tissue regeneration[18]. Indeed, we have previously described that in the SVZ the paternally expressed gene Delta-like homolog 1 (*Dlk1*), an atypical Notch ligand, plays a relevant function in postnatal neurogenesis. *Dlk1* is canonically imprinted elsewhere in the brain, however, it shows a selective absence of imprinting in subventricular NSCs with biallelic expression being required for stem cell maintenance and, ultimately, neurogenesis to the OB[16]. Another imprinted gene, the Insulin-like growth factor2 (*Igf2*), which is canonically expressed from the paternally inherited allele, is biallelically expressed in the choroid plexus resulting in a higher dose of IG2 secreted into the cerebrospinal fluid to regulate NSC proliferation[17,19]. Therefore, determining how imprinted genes operate in concert with signaling cues, as well as discovering the factors that modify methylation at imprinted regions in adult NSCs of different neurogenic niches, will lead to a better understanding of adult neurogenesis.

Genomic imprinting is regulated by epigenetic mechanisms, in particular DNA methylation imprints, that establish and maintain parental identity[20]. Epigenetic mechanisms are heritable modifications to DNA and chromatin, that do not involve changes in the DNA sequence itself, but which can modulate gene expression

and genome function[21]. At the molecular level, epigenetic regulation involves chemical modifications to DNA such as methylation and hydroxymethylation, and to the histone proteins around which the DNA is wrapped, having a relevant role in neurogenesis[22]. Ten-eleven translocation (TET) family of enzymes have been found to play an important role in passive replication-dependent and active demethylation of DNA in mammals[23]. These processes can involve the 5-hydroxymethylcytosine (5hmC), 5-formylcytosine (5fC), and 5-carboxylcytosine (5caC) intermediates that are produced by the oxidation of 5-methylcytosine (5mC)[24,25]. In mammals, three members of the TET family have been identified: TET1, TET2, and TET3[24]. Loss of TET enzymes revealed promoter hypermethylation and deregulation of genes implicated in embryonic development and differentiation[26]. Specifically, elimination of *Tet3* results in defects in neural progenitor proliferation and differentiation[27–29]. In line with this, while *Tet3* knockout ESCs can be successfully induced to neural progenitors, they undergo rapid apoptosis and show greatly compromised terminal differentiation[30].

TET proteins have been implicated in maintaining DNA methylation at imprinted regions during the germline resetting of genomic imprints and in embryonic stem cells[31–33]. However, whether TET proteins can also be involved in the modulation of DNA imprints in adult NSCs has not been established. In addition to the well-described catalytic activity of TET proteins, the preferential binding of TET proteins at 5mC-free promoters and their interaction with histone deacetylases, acetyltransferases, Polycomb repressive complex 2, and O-linked N-acetylglucosamine transferase to regulate gene expression[34,35], suggest additional potential functions independent of its catalytic activity[36]. This possibility is supported by the demonstration that non-catalytic forms of TET are able to rescue the proliferative phenotype of *Tet2* knockout cells[37]. Moreover, overexpression of TET1 or a catalytically inactive mutant in the mouse hippocampus results in the upregulation of several neuronal memory-associated genes[38], further supporting non-conventional functions of TET proteins.

There are many intriguing questions concerning the role of genomic imprinting and gene dosage during the neurogenesis process. Here we demonstrate a function for the non-catalytic form of TET3 in the maintenance of adult NSC state by preventing the differentiation of type B cells into non-neurogenic astrocytes. The mechanism involves direct binding and repression of TET3 to the imprinted gene *Snrpn*. We also show that astrocytic differentiation of adult NSCs in response to *Snrpn* deregulation involves increased production of BMP2. Our study provides important new insights into the role of TET3 in neurogenesis and emphasizes the importance of correct regulation of imprinted genes in the neurogenic niche.

## Results

**Tet3 deficiency causes depletion of the NSC pool in vivo.** TET dioxygenases convert 5mC to 5hmC resulting in the removal of the methylated cytosine in somatic tissues, including the brain[39–41]. Quantitative RT-PCR (qPCR) assays showed that *Tet2* and *Tet3* were the most abundant members of the *Tet* dioxygenases in the adult SVZ, being also highly expressed in NSCs isolated from the early postnatal SVZ (Fig. 1a). However, only *Tet3* expression was maintained postnatally in the stem cell pool and in differentiated cells (Fig. 1a). Immunofluorescent analyses with antibodies to TET3 and to cell-identity antigens in wild-type adult brains revealed nuclear staining for TET3 protein in the GFAP/SOX2+population located close to the lateral ventricles (Fig. 1b), whereas no expression was observed in

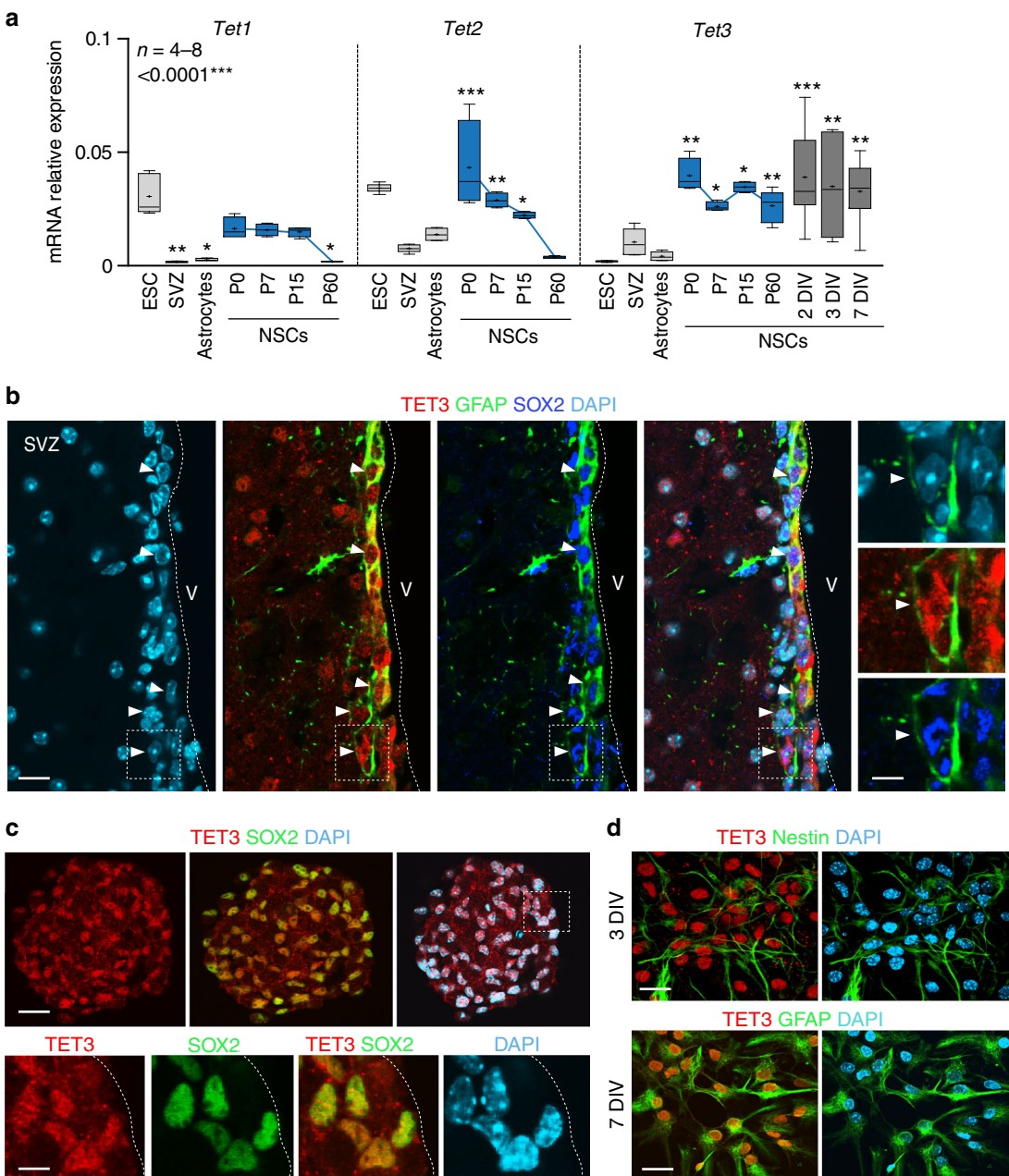

**Fig. 1** TET3 is highly expressed in adult NSCs in vivo and in vitro. **a** qPCR for *Tet1*, *Tet2* and *Tet3* genes in different cell types and tissues. *Tet1* is low expressed in adult NSCs and *Tet2* downregulates postnatally. *Tet3* is highly expressed in the NSC population and maintained postnatally. Significant levels of *Tet3* were also observed in NSCs after 2, 3 and 7 days (DIV) of differentiation in vitro. P: postnatal day. Embryonic stem cell lines (ESCs) were used as a control of expression. Data are expressed relative to *Gapdh*. Mean is indicated in the boxplots as "+". **b** Immunohistochemistry images for TET3 (red), GFAP (green), and SOX2 (blue) in the SVZ of adult wild-type mice. High-magnification images are shown. Dark arrowheads indicate triple positive astrocytes cells. V lateral ventricle lumen. **c** Immunocytochemistry for TET3 (red) and SOX2 (green) in proliferating neurospheres isolated from the adult SVZ. **d** Immunocytochemistry for TET3 (red) and Nestin (green) or GFAP (green) in adult NSCs after 3 and 7 days of differentiation. DAPI was used to counterstain DNA. One-way ANOVA and Tukey post-test were applied. All error bars show s.e.m. Number of samples used is indicated. Scale bars, 20 μm (high-magnification images, 7 μm). Source data are provided as a Source Data file

doublecortin+(DCX+) neuroblasts (Supplementary Figure 1a). TET3 staining was also observed in mature neurons of the striatal parenchyma (Supplementary Figure 1b). Furthermore, TET3 was present in all proliferating SOX2+cells in neurosphere cultures from the adult SVZ (Fig. 1c). Consistent with the analyses at the mRNA level the TET3 protein was detected in Nestin+progenitors and GFAP+astrocytes at 3 and 7 days of in vitro differentiation, respectively (Fig. 1d). Taken together, these data suggested a role for TET3 in adult NSCs.

In order to evaluate the function of TET3 in the adult SVZ, a murine genetic model was generated by crossing male mice carrying loxP sites flanking part of the *Tet3* gene (*Tet3^loxp/loxp*)[42,43] with female mice expressing the Cre-recombinase under the control of the Gfap promoter (*Gfap-cre^+/0*)[44]. Importantly, X-gal histochemistry and immunostaining for β-galactosidase in the adult brain of ROSA26R;Gfap-cre^+/0 mice (*Gfap-cre/LACZ*) and control mice (*Gfap-control/LACZ*) showed positive staining in the SVZ, RMS and olfactory bulbs of the adult *Gfap-cre/LACZ* brains

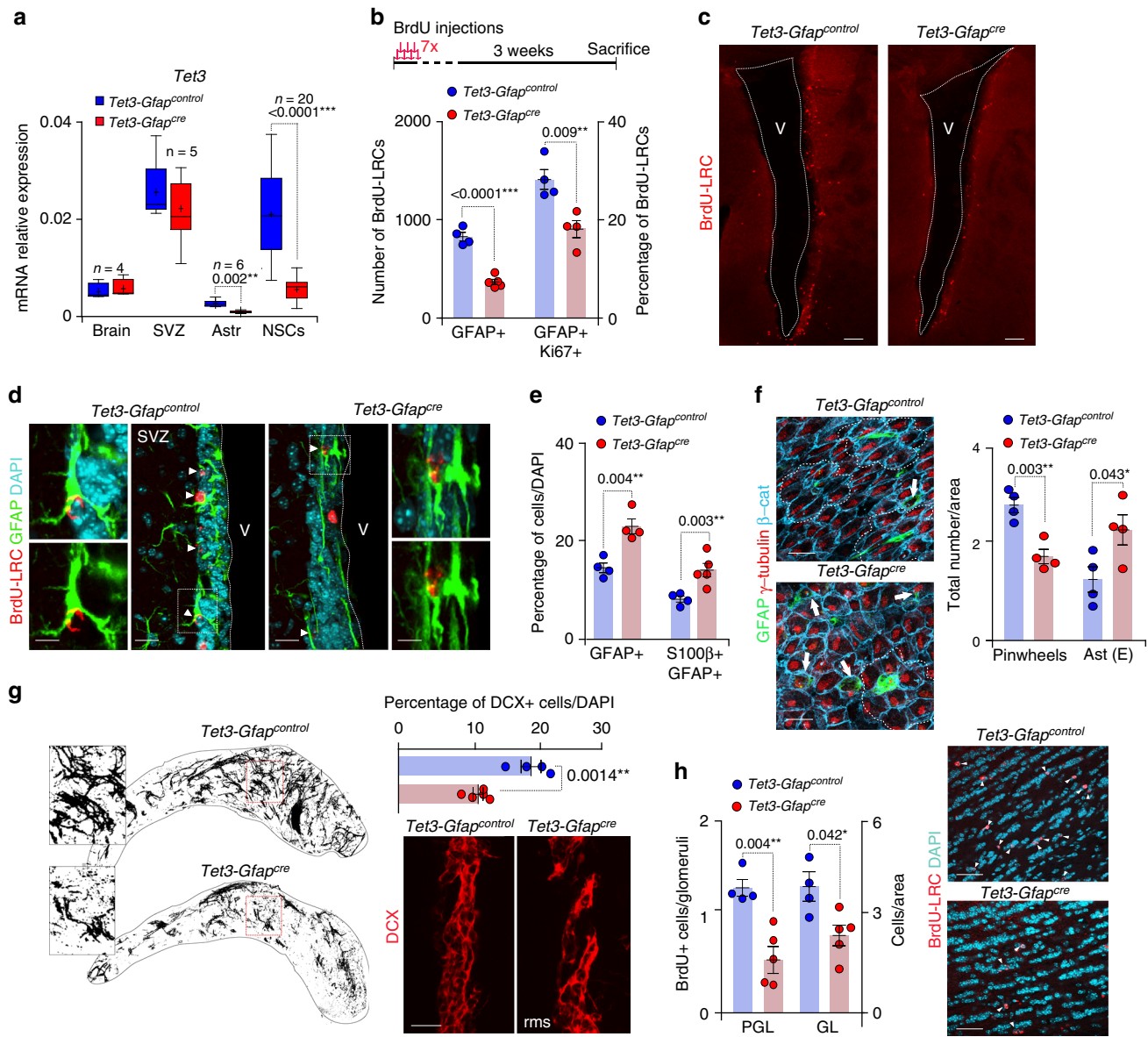

**Fig. 2** Removal of *Tet3* in the SVZ GFAP+population causes a depletion of the neural stem cell pool in vivo. **a** qPCR for *Tet3* in adult brain, SVZ, astrocytes and NSCs from *Tet3-Gfap^control* and *Tet3-Gfap^cre* mice. Data are expressed relative to *Gapdh*. Mean is indicated in the boxplots as "+". **b** Schematic drawing of the BrdU injection protocol (upper panel). Number of BrdU-label-retaining cells (BrdU-LRCs) that are GFAP+and GFAP/Ki67+in the SVZ of *Tet3-Gfap^control* and *Tet3-Gfap^cre* mice (lower panel). **c** Immunohistochemistry panoramic images of *Tet3-Gfap^control* and *Tet3-Gfap^cre* SVZ stained for BrdU label-retaining cells (LRCs; red). **d** Immunohistochemistry images for BrdU-LRCs (red) and GFAP (green) in the SVZ of mice from both genotypes. Double-positive cells are indicated with arrowheads. **e** Percentage of GFAP+and S100β/GFAP+astrocytes cells in the SVZ of *Tet3-Gfap^control* and *Tet3-Gfap^cre* mice. **f** Immunohistochemistry images for GFAP (green), γ-tubulin (red) and β-catenin (blue) in SVZ whole-mount preparations identifying pinwheel structures in both genotypes (left panels). Total number of pinwheels and differentiated astrocytes, found in the SVZ of *Tet3-Gfap^control* and *Tet3-Gfap^cre* mice (right panels). **g** Confocal reconstructions of whole-mount staining for the neuroblast marker DCX in the rostral migratory stream (RMS) of mice from both genotypes. Inserts show high-magnification images of DCX+chains (left panels). Graph shows the quantification of the percentage of DCX+cells. Immunohistochemistry images for DCX (red) in the SVZ of *Tet3-Gfap^control* and *Tet3-Gfap^cre* mice (right panels). **h** Quantification of the number of newborn neurons incorporating in the granular (GL) and periglomerular (PGL) layers in the olfactory bulbs (OB) of *Tet3-Gfap^control* and *Tet3-Gfap^cre* mice (left panel). Immunohistochemistry images for BrdU-LRCs (red) in the granular layer of the OBs from both genotypes. V lateral ventricle lumen. DAPI was used to counterstain DNA. All error bars show s.e.m. Unpaired two-tailed Student's *t*-test was used. *P*-values and number of mice used are indicated. Scale bars in (**c**), 100 μm; in (**d** and **g**), 30 μm (inserts in (**d**), 7 μm); in (**f**), 15 μm; in (**h**), 40 μm. Source data are provided as a Source Data file

(Supplementary Figure 2a) and colocalized with GFAP+cells (Supplementary Figures 2b,c), corroborating the specific deletion of *Tet3* in the adult GFAP+stem cell population. Significant expression of Cre-recombinase in *Tet3^loxp/loxp-Gfap-cre^+/0* (*Tet3-Gfap^cre*) compared to Tet3^loxp/loxp-Gfap-cre^0/0 (*Tet3-Gfap^control*) control mice was confirmed in SVZ-derived NSCs (Supplementary

Figure 2d). Cre expression in NSCs isolated from *Tet3-Gfap^cre* deficient SVZ correlated with reductions in *Tet3* mRNA and protein without compensatory changes in *Tet1* and *Tet2* (Fig. 2a and Supplementary Figures 2e–g). As expected, body and brain weights were not affected in *Tet3-Gfap^cre* mice compared to controls (Supplementary Figures 2h).

Two-month-old *Tet3-Gfap*[cre] and *Tet3-Gfap*[control] mice were injected with the nucleoside analog BrdU three weeks before sacrifice (Fig. 2b). BrdU is specifically retained in some slowly proliferating NSCs (label-retaining cells, LRCs) and in OB newborn neurons that ceased to divide and underwent terminal differentiation soon after the injection[45]. The number of GFAP +LRCs was significantly reduced in TET3-deficient mice (Fig. 2b-d and Supplementary Figure 3a) and fewer of them were positive for the proliferation antigen Ki67 (Fig. 2b and Supplementary Figure 3b), suggesting a role for TET3 in regulating the number of activated NSCs within the SVZ. Conversely, related to the decrease in activated NSCs in *Tet3*-deficient mice, we found a higher proportion of GFAP+cells with detectable levels of the calcium-binding protein S100β, a marker of terminally differentiated astrocytes (Fig. 2e). Type B cells are elongated and bear an apical process that extends through the cells of the ependymal monolayer, to end in a primary cilium immersed in the cerebrospinal fluid[8]. Monociliated B cells form structures known as pinwheels, and can, therefore, be readily identified in whole-mount preparations of the SVZ, following immunofluorescence for GFAP and the basal body marker γ-tubulin. In line with the previous findings, we observed a decrease in the number of GFAP/γ-tubulin+Type B cells contacting the ventricle in whole-mounts of the *Tet3-Gfap*[cre] SVZ (Fig. 2f). Interestingly, we also found increased proportions of monociliated GFAP+cells with cell bodies completely intercalated within the ependymal layer, a feature that has been reported to increase with aging[46] (Fig. 2f). Less densely populated DCX+neuroblast chains in the RMS (Fig. 2g) and fewer BrdU+newly-generated neurons in the granular and periglomerular layers of the OB were found in *Tet3-Gfap*[cre] mice (Fig. 2h) likely as a result of the reduced number of NSCs in the TET3 mutant SVZ. However, no changes were observed in the density of newly-formed oligodendrocytes, scored as BrdU-LRC/OLIG2+cells in the *corpus callosum* (CC) of *Tet3*-deficient mice (Supplementary Figure 3c). These data together indicate that TET3 is required for the maintenance of the NSC pool in the SVZ of adult mouse brains by preventing their premature differentiation into terminally differentiated astrocytes thus limiting neurogenesis in vivo.

***Tet3* is required to maintain self-renewal of adult NSCs.** Individual cells dissected from the postnatal SVZ can proliferate in medium containing basic fibroblast growth factor (FGF2) and/or epidermal growth factor (EGF) to produce multipotent clonal aggregates, called neurospheres[45,47]. To investigate cell-intrinsic properties of TET3, we derived neurospheres from adult *Tet3-Gfap*[cre] SVZ under controlled culture conditions (Fig. 3a and Supplementary Figure 4a). Expression of cre-recombinase and downregulation of *Tet3* was again confirmed in cultured neurospheres (Fig. 2a and Supplementary Figures 2d, e). SVZ isolated from *Tet3-Gfap*[cre] adult mice yielded fewer primary neurospheres compared to those of controls (Fig. 3b). To assess specifically the self-renewal capability of these cultures, primary neurospheres were individually dissociated into single cells and plated at limiting dilution (2.5 cells/μl) for several passages (Fig. 3a). *Tet3*-deficient neurospheres produced a significantly reduced number of new secondary clones at different passages (Fig. 3b, c). In contrast, TET3 did not appear to regulate proliferation, as we found no changes in diameter in primary or secondary *Tet3*-deficient neurospheres (Fig. 3d). A cell cycle analysis of *Tet3-Gfap*[control] and *Tet3-Gfap*[cre] neurosphere cultures revealed no changes in the proportion of cells in different cell cycle phases (Fig. 3e). The reduction in the number of neurospheres could not be ascribed to a decrease in cell survival as evidenced by similarly low proportions of caspase 3+apoptotic cells in neurosphere

cultures from the two genotypes (Fig. 3e and Supplementary Figure 4b). As a result of the self-renewal defect, the bulk growth expansion rate of *Tet3-Gfap*[cre] cultures after several passages was impaired compared to the *Tet3-Gfap*[control] cells (Fig. 3f). These findings indicate that TET3 sustains self-renewal without affecting the overall proliferation rate or cell survival of adult NSCs.

**TET3 prevents terminal astrocytic differentiation of NSCs.** Adult NSCs are multipotent cells that can form neurons, astrocytes, and oligodendrocytes ex vivo. Indeed, when neurosphere cultures are induced to differentiate in vitro they undergo an orderly series of intermediate steps of proliferation and cell fate restriction resembling those occurring in vivo[48,49] and resulting in the differentiation of derived neural progenitors into different neural cell types (Supplementary Figure 4a). Because the in vivo data suggested that *Tet3* deficiency resulted in increased astrocytic differentiation, we monitored the in vitro behavior of neural progenitors derived from *Tet3-Gfap*[cre] and control neurospheres plated under adherent conditions in the absence of mitogens. During the first 2 days (2-DIV) of differentiation, a decrease in the neural progenitor marker Nestin together with an increase in the terminally differentiated astrocytic marker *S100β* was observed in *Tet3-Gfap*[cre] compared to control cultures (Fig. 3g), whereas no change in the levels of expression of the neuronal βIII-tubulin gene (*Tubb3*) was found (Supplementary Figure 4c). At the protein level, the deficiency in TET3 resulted in reduced proportions of Nestin+cells after 2-DIV of differentiation (Fig. 3h, i). Higher proportions of cells that were strongly positive for S100β, a protein largely absent from neurogenic GFAP+cells correlating with loss of neurosphere-forming potential[50], was also observed after 7-DIV of differentiation (Fig. 3h, i). Conversely, neuronal and oligodendroglial differentiation was normal as no significant differences were found in the percentage of βIII-tubulin+or O4+cells, respectively (Supplementary Figures 4d,e).

In order to test whether enhanced astrocytic differentiation in the absence of *Tet3* was consistently accompanied by a reduction in the capacity of NSCs to form neurospheres, 7-DIV differentiated NSCs cultures were detached and replated under proliferating conditions with mitogens (Supplementary Figure 4a). This led to the reactivation of a small proportion of cells with the capacity to form neurospheres in non-adherent conditions after 12 days in vitro (12 DIV) (Supplementary Figure 4f). The number of neurosphere-forming cells in *Tet3-Gfap*[cre] NSCs was significantly reduced, indicating that the bias toward a more differentiated astrocytic phenotype in the absence of TET3 correlated with a reduction in stemness (Supplementary Figure 4f). This suggests that TET3 can directly promote the neurogenic potential of the multipotential stem cell-like astrocytes by preventing their premature differentiation.

**TET3 regulates NSCs maintenance through repression of *Snrpn*.** To understand the molecular mechanism underlying the differentiation defects observed in *Tet3*-deficient NSCs, a RNA-seq was performed in control and *Tet3-Gfap*[cre] neurosphere cultures. RNA-seq analysis confirmed the downregulation of *Tet3* mRNA levels in *Tet3-Gfap*[cre] compared to control cultures (LogFC in *Tet3-Gfap*[cre] neurosphere cultures: $-1.0074$ with a FDR $= 3.97 \times 10^{17}$). *Tet3* depletion resulted in 97 significantly downregulated and 96 upregulated genes (Fig. 4a; Supplementary Figure 5a; Supplementary Data set 1 and 2). Interestingly, Gene Set Enrichment Analysis (GSEA) revealed specific changes in pathways promoting cell growth such as PI3K-Akt, MAPK, and Wnt signaling pathways and genes involved in synaptic plasticity in *Tet3-Gfap*[cre] compared to *Tet3-Gfap*[control] NSCs (Supplementary Figure 5b).

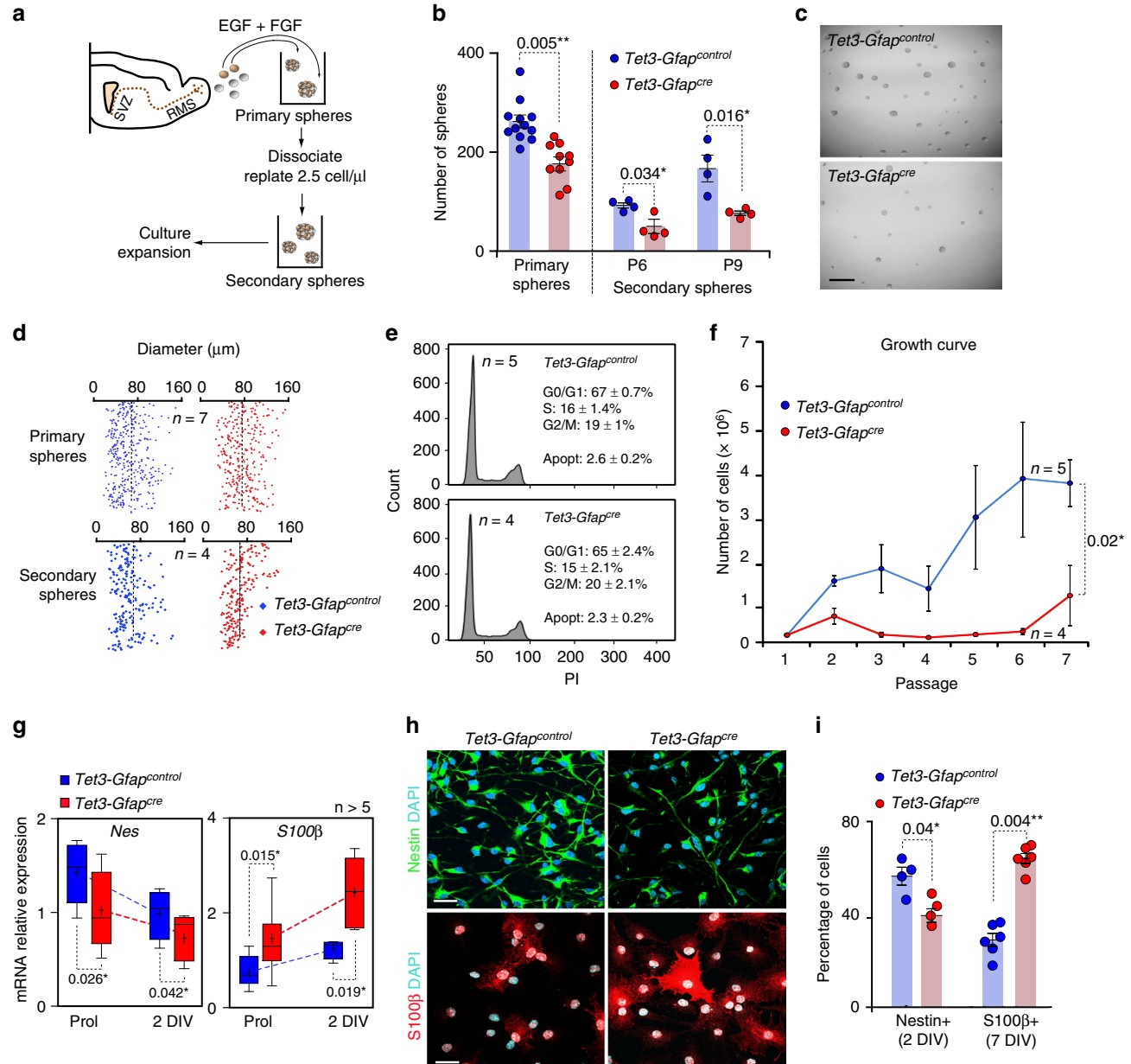

**Fig. 3** *Tet3* deficiency causes a decrease in NSCs self-renewal triggering their terminal differentiation into non-neurogenic astrocytes. **a** Schematic representing the primary and secondary forming neurosphere assay and the expansion protocol. Neurospheres are cultured in proliferation conditions with the mitogens EGF and FGF. Cells are dissociated and replated at 2.5 cells/μl. Secondary neurospheres are counted 5 days later. **b** Number of primary neurospheres obtained from the SVZ of *Tet3-Gfap^control* and *Tet3-Gfap^cre* mice. Number of secondary spheres formed after passages (P) 6 and 9 is also shown. **c** Representative images of secondary neurosphere cultures from both genotypes. **d** Diameter of primary and secondary neurospheres formed in *Tet3-Gfap^control* and *Tet3-Gfap^cre* cultures. Dashed lines indicate the mean diameter. **e** Cell cycle profiles in neurosphere cultures from both genotypes after 3 days in vitro. Percentages of G0/G1, S and G2/M phases are indicated. Percentages of apoptosis are also shown. **f** Growth curve showing the total number of cells formed after 7 passages in *Tet3-Gfap^control* and *Tet3-Gfap^cre* neurosphere cultures. **g** qPCR for the undifferentiation marker *Nestin* and the terminally differentiated marker *S100β* in *Tet3-Gfap^control* and *Tet3-Gfap^cre* NSCs in proliferation conditions (Prol) and after 2 days of differentiation (2 DIV). Data are expressed relative to *Gapdh*. Mean is indicated in the boxplots as "+". **h** Immunocytochemistry images for Nestin (green; upper panels) and S100β (red; lower panels) in NSCs of both genotypes after 2 or 7 days of differentiation, respectively. DAPI was used to counterstain DNA. **i** Percentage of cells that are positive for Nestin and S100β in *Tet3-Gfap^control* and *Tet3-Gfap^cre* NSCs after 2 or 7-DIV of differentiation, respectively. All error bars show s.e.m. Unpaired two-tailed Student's *t*-test was used. *P*-values and number of samples are indicated. Scale bars in (**c**), 200 μm and in (**h**), 40 μm. Source data are provided as a Source Data file

Non-canonical regulation of imprinted genes has been shown to regulate adult neurogenesis[16,17,51]. Moreover, the brain expresses high levels of 5hmC, suggesting a potential contribution of TET3 to the epigenetic state at imprinted regions in adult NSCs. Based on the RNA-seq datasets, we next focused on changes in expression of all known imprinted genes (Supplementary Dataset 3). From around 150 imprinted genes analyzed, three showed a significant change in mRNA expression. *Cntn3* (Contactin 3) was downregulated in *Tet3-Gfap^cre* NSCs, whereas *Cobl* (Cordon-bleu WH2 repeat) and *Snrpn* (Small nuclear

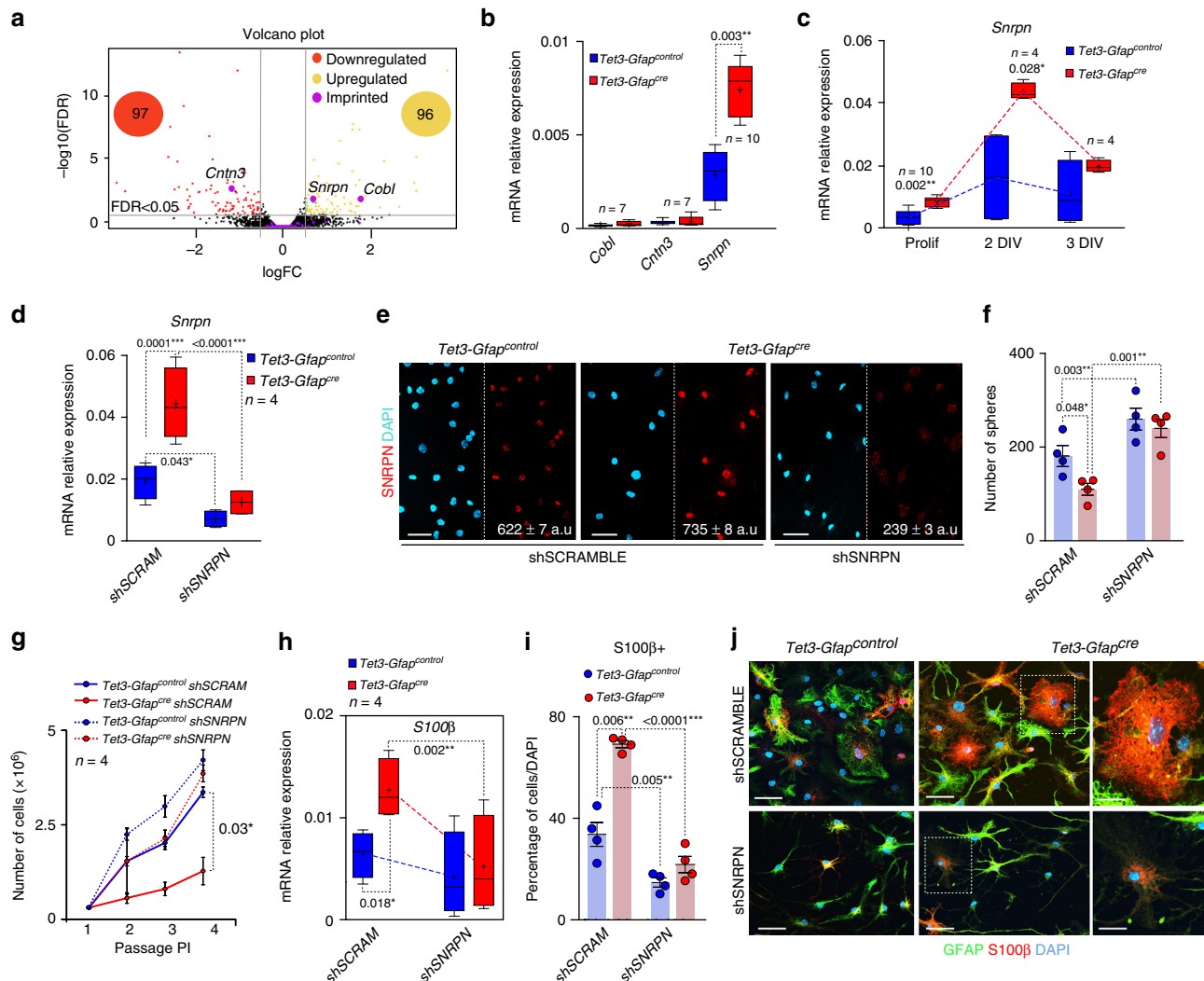

**Fig. 4** Depletion of *Tet3* in NSCs increases the expression of the imprinted gene *Snrpn*. **a** Volcano plot for genes differentially expressed by RNAseq in *Tet3-Gfap*<sup>cre</sup> compared to control NSCs. Downregulated genes are in red and upregulated genes in yellow. Imprinting genes are showed in purple. **b** qPCR for *Cntn3, Cobl,* and *Snrpn* in NSCs from both genotypes. **c** qPCR for *Snrpn* in NSCs from both genotypes in proliferating conditions and after 2 or 3 days of differentiation. The pick of expression of the gene occurs after 2 DIV of differentiation. **d** qPCR for *Snrpn* in *Tet3-Gfap*<sup>control</sup> and *Tet3-Gfap*<sup>cre</sup> NSCs that have been lentivirus transduced with a shRNA for SNRPN. A shRNA SCRAMBLE was used as a control. **e** Immunocytochemistry images for SNPRN (red) in *Tet3-Gfap*<sup>control</sup> and *Tet3-Gfap*<sup>cre</sup> NSCs 7 days after the shRNA experiment. **f** Number of secondary spheres formed after *Tet3* downregulation in *Tet3-Gfap*<sup>cre</sup> cultures. **g** Growth curves showing the total number of cells formed after 4 passages in *Tet3-Gfap*<sup>control</sup> and *Tet3-Gfap*<sup>cre</sup> neurosphere cultures that have been interfered with the shRNA for *Snrpn*. **h** qPCR for the astrocytic differentiation marker S100β after 2 DIV of differentiation in NSCs that have been lentivirus transduced with a shRNA for SNRPN. **i** Percentage of S100β+cells (relative to total number of DAPI) in shRNA interfered *Tet3-Gfap*<sup>control</sup> and *Tet3-Gfap*<sup>cre</sup> cultures. **j** Immunocytochemistry images for S100β (red) and GFAP (green) in interfered NSCs of both genotypes after 7-DIV of differentiation. Data are expressed relative to *Gapdh*. DAPI was used to counterstain DNA. All error bars show s.e.m. One-way ANOVA and Tukey post-test were applied. *P*-values and number of samples are indicated. Mean is indicated in the box and whiskers plots as "+". Scale bars in (**e** and **j**), 40 μm (high-magnification images 15 μm)

ribonucleoprotein-associated polypeptide N) showed increased levels of expression in *Tet3*-deficient neurospheres compared to controls (Fig. 4a). Validation of the RNAseq data by qPCR showed very low levels of expression of *Cntn3* and *Cobl* in NSCs but confirmed the significant upregulation of *Snrpn* in *Tet3-Gfap*<sup>cre</sup> neurospheres (Fig. 4b; Supplementary Figure 6a). A more detailed analysis of *Snrpn* expression in NSCs revealed that the gene was significantly expressed both in proliferating and differentiating conditions with a peak of expression at the first steps of the process of differentiation (Fig. 4c). Accordingly, although *Snrpn* was increased in *Tet3-Gfap*<sup>cre</sup> compared to *Tet3-Gfap*<sup>control</sup> proliferating NSCs, the maximum difference of *Snrpn*

mRNA levels in both genotypes was observed after 2-DIV of differentiation (Fig. 4c).

In order to determine whether the augmented levels of *Snrpn* in *Tet3*-deficient NSCs might be functionally responsible for the loss of stemness and enhanced astrocytic differentiation of NSCs, a lentivirus (Lv)-mediated short hairpin (sh) RNA was used to silence *Snrpn* (shSNRPN) expression in *Tet3-Gfap*<sup>cre</sup> neurosphere cultures. Downregulation of *Snrpn* was verified by qPCR and immunocytochemistry in transduced cells (Fig. 4d, e) and resulted in the reversion of the neurosphere-forming capacity of *Tet3*-deficient cells to wild-type levels (Fig. 4f and Supplementary Figure 6b). Accordingly, the growth expansion rate of

NSCs was increased to normal in *Tet3-Gfap^cre* neurosphere cultures infected with the shSNRPN (Fig. 4g). To assess whether this rescue in the self-renewal and expansion capacities in *Tet3*-deficient cultures might be due to an independent effect of *Snrpn* on proliferation, neurospheres size was determined, however no changes were found in mean diameter in any condition (Supplementary Figures 6b, c) suggesting a functional role of *Snrpn* in NSCs self-renewal. Consistently, lentiviral delivery of the shSNRPN in *Tet3*-deficient cells restored S100β levels to normal after 2 and 7-DIV of differentiation (Fig. 4h and Supplementary Figure 6d). Moreover, the percentage of S100β +cells in *Tet3-Gfap^cre* differentiated cultures was restored to wild-type levels (Fig. 4i, j). No changes in the levels of *Tubb3* mRNA or in the proportion of βIII-tubulin+cells were observed after knockdown of *Snrpn* levels in *Tet3-Gfap^cre* neurospheres (Supplementary Figures 6d,e). Notably, downregulation of *Snrpn* in *Tet3-Gfap^control* cells, caused an increase in neurosphere formation and cell growth and decreased the proportion of differentiated S100β+cells (Fig. 4d–j). These data suggested that TET3 directly promotes the neurogenic potential of multipotent stem cell-like astrocytes via regulation of *Snrpn*, antagonizing their premature terminal differentiation into mature astrocytes.

**Snrpn controls BMP to mediate differentiation of NSCs.** Because our data indicated that *Tet3* deficiency results in increased astrogliogenesis, we focused on Bone Morphogenetic Proteins (BMPs), cytokines that are strong promoters of gliogenesis in vivo and in vitro[10,52]. Real-time PCR analysis of the expression of these molecules revealed an enrichment of *Bmp2* mRNA in the absence of TET3 that was completely restored to normal levels by interference of the *Snrpn* gene (Fig. 5a). BMP ligands signal through a tetrameric complex that is formed by BMP type II receptor (BMPR-II) and different classes of BMP type I receptors. Activation of BMPR-IA or -IB results in the phosphorylation of SMAD1, SMAD5, and SMAD8 DNA binding proteins, which heterodimerize with co-SMAD4, and together translocate to the nucleus to regulate gene expression[53]. To test whether activation of the pathway in response to elevated levels

of BMP2 was increased in *Tet3*-deficient NSCs, we used pan-specific antibodies to pSMAD1 and pSMAD5 (pSMAD1/5). Treatment of wild-type NSCs with 20 ng/ml BMP2 resulted in increased immunofluorescent detection of nuclear pSMAD and premature differentiation of S100β+astrocytes (Supplementary Figures 6f–h), indicating that differentiating NSCs are responsive to BMPs. Notably, increased levels of nuclear pSMAD1/5 were detected in *Tet3-Gfap^cre* compared to *Tet3-Gfap^control* NSCs (Fig. 5b). Furthermore, this was antagonized by the addition of Noggin, a natural antagonist of BMP and by the downregulation of the *Snrpn* gene (Fig. 5b).

To further substantiate these data, we used a luciferase reporter construct that contains two copies of two distinct highly conserved BMP-responsive elements (BREs) and the associated regulatory motifs of the natural human and mouse Id1 promoter as a pSMAD activity sensor (BRE-tk-luciferase)[54]. Consistent with the data described above, we found an increase in the basal activity of the BRE reporter in *Tet3-Gfap^cre* when compared with wild-type cells, and this increase was efficiently abrogated by the addition of Noggin, as well as by the interference of *Snrpn* (Fig. 5c). Taken together, these results demonstrate that BMP signaling is augmented in the absence of *Tet3* and that control of BMP2 by SNRPN is functionally responsible for the enhanced astrocytic differentiation of *Tet3*-deficient NSCs.

**A non-catalytic action of TET3 contributes to Snrpn repression.** Significant levels of 5mC and 5hmC were found by immunocytochemistry in neurospheres isolated from the adult SVZ (Supplementary Figure 7a). Analysis of 5hmC and 5mC staining in vivo also showed variable intensities between different cellular populations within the adult SVZ (Supplementary Figures 7b, c). In particular, distribution of 5hmC and 5mC showed high levels of 5hmC and low levels of 5mC in the GFAP/SOX2 +stem cell population (Supplementary Figures 7b, c). In contrast, no 5hmC was found in the more differentiated cells such as the doublecortin (DCX)+neuroblast population (Supplementary Figure 7b). Thus, the presence of 5hmC in the NSC population along with its apparent exclusion from differentiated cells,

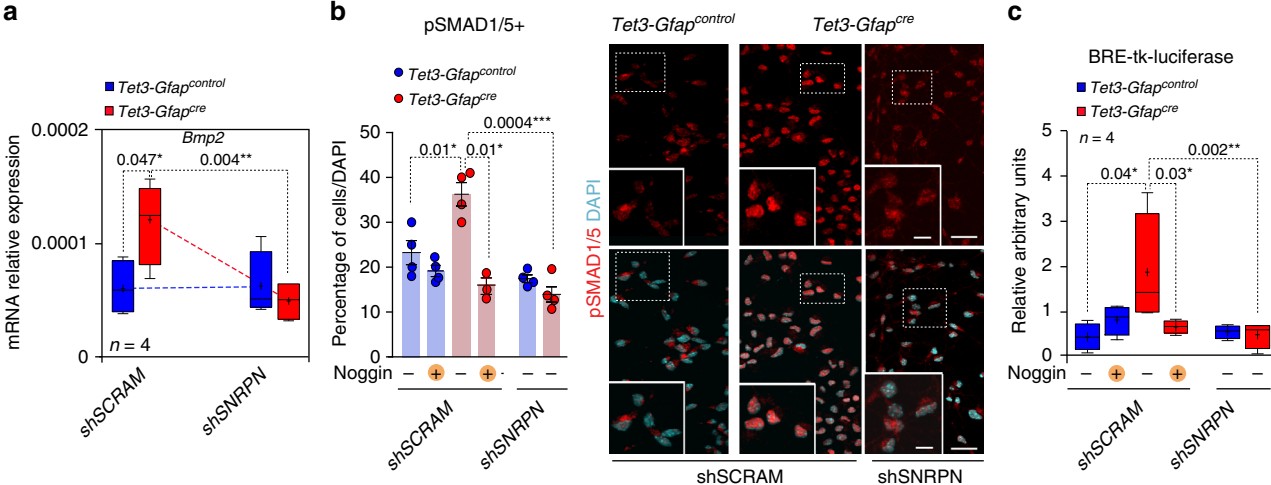

**Fig. 5** BMP2 mediates terminal differentiation of *Tet3*-deficient NSCs. **a** qPCR for *Bmp2* in *Tet3-Gfap^control* and *Tet3-Gfap^cre* interfered NSCs after 2 DIV of differentiation. Data are expressed relative to *Gapdh*. **b** Percentage of pSMAD1/5+cells relative to total cells in *Tet3-Gfap^control* and *Tet3-Gfap^cre* NSCs that have been interfered with a shRNA for SNRPN (left). NSCs were treated with 100 ng/ml of the BMP natural antagonist Noggin. Immunocytochemistry images for pSMAD1/5+cells (right). DAPI was used to counterstain DNA. **c** BRE-tk-luciferase reporter activity in *Tet3-Gfap^control* and *Tet3-Gfap^cre* shRNA interfered neurospheres and treated with Noggin. All error bars show s.e.m. One-way ANOVA and Tukey post-test were applied. *P*-values and number of samples are indicated. Mean is indicated in the box and whiskers plots as "+". Scale bars in (**b**), 40 μm (high-magnification images 15 μm). Source data are provided as a Source Data file

suggested that formation of 5hmC might also participate in the function and/or maintenance of the undifferentiated state in the adult neurogenic niches. Notably, upon *Tet3* knockdown no changes in global 5mC or 5hmC levels, determined by ELISA, were observed in adult *Tet3-Gfap^control* and *Tet3-Gfap^cre* neurosphere cultures (Supplementary Figure 7d). Consistently, immunohistochemistry for 5hmC within the *Tet3-Gfap^cre* SVZ revealed no significant changes in global 5hmC in *Tet3*-deficient NSCs in vivo (Supplementary Figure 7c).

*Snrpn* belongs to the cluster of imprinted genes associated with Prader–Willi Syndrome and is canonically expressed from the paternally inherited chromosome[55]. The mouse *Snrpn* gene has a germline derived differentially methylated region (DMR) that is associated with imprinted *Snrpn* expression, and is required for imprinting control of the domain (Supplementary Figure 8a)[56,57]. We have previously shown the expected paternal expression of *Snrpn* in normal tissue and adult NSCs[16], but loss of imprinting of the gene could explain the upregulation of *Snrpn* in *Tet3-Gfap^cre* NSCs. To test this, bisulfite sequencing of the *Snrpn* DMR was performed in *Tet3-Gfap^control* and *Tet3-Gfap^cre* neurosphere cultures (Supplementary Figure 8a, b). Both control and mutant neurospheres showed 50% methylation indicating retention of imprinting. Bisulfite sequencing cannot distinguish between 5mC and 5hmC; therefore, to specifically assess 5hmC levels at the *Snrpn* DMR, a hydroxymethylation immunoprecipitation using antibodies against 5hmC (hydroxyMeDIP) was developed in *Tet3-Gfap^control* and *Tet3-Gfap^cre* NSCs cultures. Consistent with the bisulfite sequencing analysis, no differences in 5hmC levels were observed at the *Snrpn* promoter (Supplementary Figure 8c), suggesting that there was no loss of imprinting of the *Snrpn* gene.

The remarkable increase of *Snrpn* expression independently of a change to its DMR methylation and hydroxymethylation in *Tet3*-deficient NSCs, prompted us to explore the possibility that TET3 might have non-catalytic functions in NSCs. We first used chromatin immunoprecipitation followed by qPCR (ChIP-qPCR) to assess direct binding of TET3 to the *Snrpn* promoter in wild-type adult NSCs. Two *Snrpn* promoter regions (R1, −20 to +132 bp, and R2, +271 to +434 bp) and *Clcn6* promoter, as a positive control region[58], were chosen for analysis (Supplementary Figure 8a). Importantly, TET3 binding was observed at the *Snrpn* promoter regions but not at a more distal upstream region (−135 kb, Region 3, R3) (Fig. 6a). To specifically determine whether TET3 was bound to the paternal transcribed (unmethylated), maternal repressed (methylated) allele or to both alleles (Supplementary Figure 8a), a ChIP-qPCR for TET3 was performed in NSCs derived from adult F1 mice hybrids offspring from *Mus musculus domesticus* (C57BL6/J) females and *Mus musculus castaneus* (CAST/EiJ) males (BxC hybrids NSCs), in which two single-nucleotide polymorphisms (SNPs) were identified between the two subspecies at the *Snrpn* promoter (Supplementary Figure 8d). Direct sequencing of these SNPs after ChIP showed preferential binding for TET3 to the paternal transcribed and unmethylated allele (Fig. 6b), supporting a methylation-independent function of TET3 on the *Snrpn* gene.

To further determine whether TET3 contributes to transcriptional fine-tuning of *Snrpn*, we overexpressed TET3 in *Tet3-Gfap^cre* NSCs and found that this caused a significant decrease in *Snrpn* expression but not in other imprinted genes such as *Mcts2* (Fig. 6c and Supplementary Figure 9a). Notably, when we overexpressed a catalytically inactive variant of TET3 (*Tet3* CDmut) we observed a comparable decrease in the levels of the *Snrpn* and *Bmp2* genes in *Tet3-Gfap^cre* NSCs (Fig. 6c). In line with this, overexpression of either wild type or catalytically inactive TET3 led to an increase in the number of spheres formed in self-renewal conditions in *Tet3-Gfap^cre* NSCs (Fig. 6d). Consistently, overexpression of TET3 in *Tet3*-deficient cells

restored the percentage of S100β+cells in *Tet3-Gfap^cre* differentiated cultures to wild-type levels (Fig. 6e, f). Overexpression of TET3 in *Tet3-Gfap^control* NSCs did not modify *Snrpn* and *Bmp2* levels (Supplementary Figure 9b). Moreover, self-renewal and astrocytic differentiation were also normal in control cultures (Supplementary Figures 9c-e). However, when we overexpressed TET3 or its catalytically inactive variant in *Tet3-Gfap^cre* cultures, we observed a decrease in the levels of pSMAD1/5 and in the basal activity of the BRE reporter (Fig. 6g, h). These data together demonstrate that TET3 binds to the *Snrpn* promoter preferentially at the paternal allele to contribute to transcriptional repression of the gene in adult NSCs, independently of its catalytic function and that BMP2 signaling is responsible for the astrocytic terminal differentiation mediated by SNRPN (Supplementary Figure 10).

## Discussion

The formation of new neurons in the adult brain throughout life involves the activation of pools of NSCs while preserving the stem cell number. Emerging evidence indicates that imprinted genes regulate the life-long function of adult NSCs[16,17,51]. This study demonstrates a role for the dioxygenase TET3 in the self-renewal and hence maintenance of NSCs within the adult SVZ. We show that loss of *Tet3* results in the differentiation of neurogenic progenitors both in vivo and in vitro and compromises the maintenance of the NSC pool in this niche leading to reduced OB neurogenesis. More interestingly, our work indicates that TET3 regulates self-renewal by repressing transcription of the imprinted gene *Snrpn* through mechanisms that are independent of its catalytic function in modulating methylation.

TET3 deficiency causes an active apoptosis of neural progenitors derived from ESCs, which results in a reduction of neuronal production[30]. This context differs from ours, as in adult NSCs *Tet3* deficiency does not modify apoptosis or survival rates. Rather, when TET3 is absent in the GFAP+population in vivo, a depletion of the adult NSC pool is observed, and this is due to their terminal differentiation into non-neurogenic astrocytes. As a consequence of the continuous depletion of the NSC pool in *Tet3*-deficient mice, the process of neurogenesis to the OB is impaired, a finding recapitulated in neurosphere cultures in vitro. Interestingly, no changes in the proportion of the cell cycle phases are observed in mutant cultures indicating that TET3 promotes self-renewal capacity without affecting their overall proliferation rate. Differentiation of both mutant and wild-type NSCs into the three cell lineages of the central nervous system supports premature differentiation of NSC into non-neurogenic astrocytes in *Tet3*-deficient NSCs.

Gene expression analyses have revealed the molecular bases of these phenotypic observations, with a significant upregulation of the *Snrpn* gene in *Tet3*-deficient NSCs. *Snrpn* belongs to the Prader–Willi imprinted gene cluster and is expressed from the paternally inherited chromosome[55]. Patients with Prader–Willi syndrome lack expression of *Snrpn* and exhibit neurological problems including learning difficulties and hyperphagia[59,60]. *Snrpn* encodes the RNA-associated SmN (survival motor neurons) protein implicated in pre-mRNA editing, which contributes to tissue-specific alternative splicing[61]. It is well-established that *Snrpn* is highly expressed in brain and its expression increases markedly during postnatal brain development[61]. Our data indicate that TET3 regulates *Snrpn* dosage in an imprint-independent manner and that this is responsible for the premature differentiation of NSCs that we observe. Our findings suggest that *Snrpn* overexpression influences the NSC pool in adults, a function that has not previously been associated with Prader–Willi Syndrome and which has implications for our understanding of the etiology of this disorder.

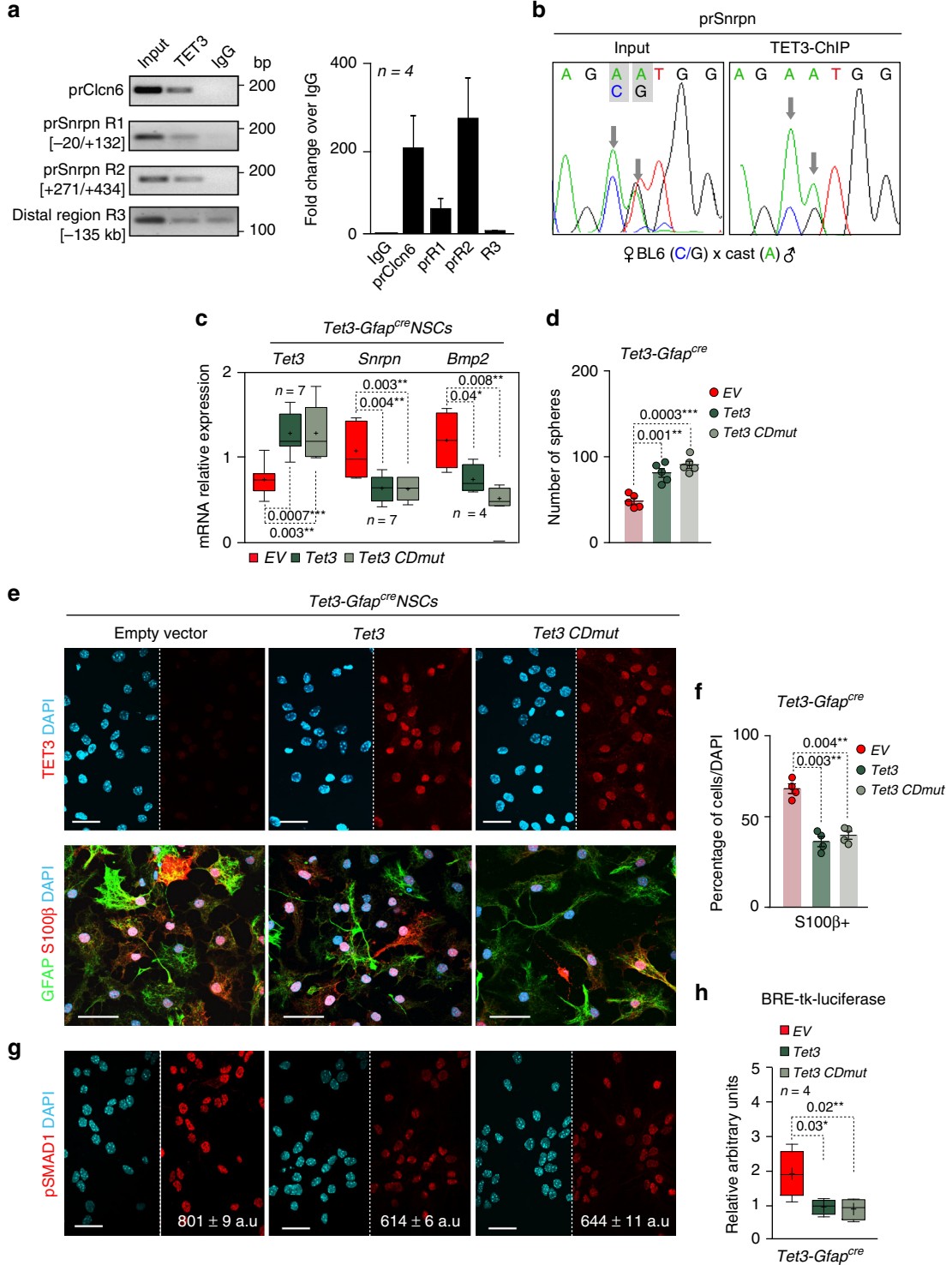

The effects of BMPs in astroglial differentiation have been reported extensively for both late fetal and postnatal progenitors[52,53], and most frequently, BMP-induced astroglial differentiation is related to deficiencies in the production of other cell lineages[53]. Moreover, BMP4 overexpression appears to accelerate the differentiation of some radial glia, suggesting that multipotential progenitors terminally differentiate when exposed to BMPs[53,62]. Our findings show that *Snrpn* overexpression correlates with elevated levels of this morphogen, suggesting that premature differentiation of *Tet3*-deficient NSCs into astrocytes may be partly

the consequence of the augmented activity of the BMP signaling. Moreover, the presence of the BMP antagonist Noggin that is produced by ependymal cells[63] could be balancing the *Snrpn* function in the SVZ, preventing the differentiation of the NSC pool. Indeed, we showed that addition of Noggin to *Tet3*-deficient cultures, restored the terminal astrocytic differentiation phenotype observed. Further work will be necessary to elucidate whether *Snrpn* has a role on the pre-mRNA editing of BMP genes.

TET enzymes regulate the balance of DNA methylation and demethylation by dynamic conversion of 5mC into 5hmC, 5fC,

**Fig. 6** TET3 binds to *Snrpn* promoter at the paternal allele to regulate its expression independently of methylation. **a** qPCR after chromatin immunoprecipitation (ChIP) of wild-type NSCs. Two regions (R1 and R2) of the *Snrpn* promoter (prSnrpn) were analyzed. *Clcn6* promoter was used as a control for TET3 binding. A non-relevant distal region (R3) was analyzed as a control of specificity. Values are shown as the fold change over the IgG. **b** DNA sequences after ChIP-qPCR for TET3 in F1 hybrid NSCs derived from *Mus musculus domesticus* (abbreviated, BL6) and *Mus musculus castaneus* (abbreviated, Cast) mice (BxC), showing the two diagnostic strain-specific polymorphisms at the *Snrpn* promoter in the input sample. A more abundant presence of the "A" nucleotide, corresponding to the paternal allele (Cast), is observed. **c** qPCR for *Tet3*, *Snrpn* and *Bmp2* genes in *Tet3-Gfap*[cre] NSCs that had been nucleofected with *Tet3* or with *Tet3* with a mutated catalytic domain (*Tet3* CDmut). An empty vector was used as a control for nucleofection. **d** Number of secondary spheres formed in *Tet3-Gfap*[cre] cultures after nucleofection with *Tet3* variants. A rescue in the self-renewal capacity was observed. **e** Immunocytochemistry images for TET3 (red) in *Tet3-Gfap*[cre] NSCs that had been nucleofected with Tet3, Tet3 CDmut or EV (upper panels). Immunocytochemistry images for GFAP (green) and S100β (red) are also shown (lower panels). **f** Percentage of S100β+cells in *Tet3-Gfap*[cre] NSCs that had been nucleofected with Tet3, Tet3 CDmut, or EV. **g** Immunocytochemistry images for pSMAD1/5+(red) in nucleofected *Tet3-Gfap*[cre] NSCs. **h** BRE-tk-luciferase reporter activity in *Tet3-Gfap*[cre] NSCs that had been nucleofected with Tet3, Tet3 CDmut, or EV. All error bars show s.e.m. One-way ANOVA and Tukey post-test were applied. *P*-values and number of samples are indicated. Mean is indicated in the box and whiskers plots as "+". Scale bars, 30 μm. Source data are provided as a Source Data file

and 5CaC[40,41]. Several other reports have shown non-catalytic functionalities of the TET proteins[23,36]. Concretely, TET3 has been implicated in the stabilization of thyroid hormones nuclear receptors promoting transcriptional activation where TET3 catalytic domain is not implicated and a TET3 catalytic dead mutant can rescue the developmental defects caused by TET3 knockdown[64]. Most notably, TET3 interacts with O-linked N-acetylglucosamine transferases (OGT) to influence transcription[34,35]. Although our data show significant amounts of 5hmC in GFAP-expressing cells within the SVZ and in proliferating cells in vitro, global levels of 5mC/5hmC are not altered in *Tet3*-deficient NSCs suggesting non-catalytic effects of TET3 in this neurogenic niche. Indeed, we demonstrate that TET3 binds to the *Snrpn* promoter at the paternal transcribed and unmethylated allele acting as a transcriptional repressor of the gene by a catalytic-independent activity mechanism, supporting a multifunctional role for TET3 in adult NSCs.

Adult niches preserve the undifferentiated property of their resident stem cells, but contribution of epigenetic mechanisms to this process has remained largely unexplored. Our data have elucidated the action of TET3, preventing terminal differentiation and exhaustion of NSCs by transcriptional repression of the *Snrpn* gene independently of its methylation. We also described how this regulation acts in concert with niche signals such as BMP and Noggin to modulate the proper differentiation of the NSC pool. Further work will be relevant to elucidate the structural, genetic and epigenetic requirements of this regulatory mechanism and its contribution to neurogenic phenotypes.

## Methods

**Animals and in vivo manipulations**. *GFAP-cre* (6.Cg-Tg(Gfap-cre)73.12Mvs/J) mice were obtained from Jackson Laboratory and genotyped as described[44]. *GFAP-cre* mice were generated using a 15 kb mouse *Gfap* promoter cassette containing all introns, promoter regulatory elements, exons, and 2 kb of 3′ and 2.5 kb of 5′ flanking regions of the mouse *Gfap* gene[44]. *Gfap* expression is prevented by the removal of a small region in exon 1. *Tet3*[loxp/loxp] mice contain LoxP sites flanking exon 5 of *Tet3* gene[43]. Expression of Cre-recombinase results in a deletion of this region and a frame-shift from exon 6 that affects all downstream exons until a premature stop codon in exon 7. To generate specific deletion of *Tet3* in GFAP positive cells, heterozygous *GFAP-cre* transgenic animals were bred to *Tet3*[loxp/loxp]. Additionally, The Jackson laboratory reports that the *GFAP-cre* line have *cre* expression in the male germline. To avoid this problem, *GFAP-cre* females and *Tet3*[loxp/loxp] males were used to generate the experimental animals. Animals were genotyped by PCR analysis of DNA, extracted from mouse ear-punch tissue with the following primers: for the presence of cre-recombinase *Cre-F* and *Cre-R* and for the presence of LoxP sites *Tet3F*, *Tet3R1*, and *Tet3R2* (Supplementary Table 1). Mice were maintained on a C57BL6 background. We have complied with ethical regulations for animal testing and research and experiments were carried out following protocols approved by the ethics committee of the University of Valencia (Spain).

**Immunohistochemistry and β-galactosidase staining**. BrdU administration regimes have been previously detailed[16,45]. Briefly: mice at 2–4-months-of-age were injected intraperitoneally with 50 mg of BrdU per kg of body weight every 2 h for

12 consecutive hours (7 injections in total). Three weeks after the injections, animals were deeply anaesthetized and transcardially perfused with 4% paraformaldehyde (PFA) in 0.1 M phosphate buffer saline pH 7.4 (PBS) and brains were vibratome-sectioned at 40 μm. For immunohistochemistry, sections were washed in PBS and blocked at room temperature for 1 h in PBS with 0.1% Triton X-100 supplemented with 10% Fetal bovine serum (FBS) and then incubated overnight at 4 °C with primary antibodies (Supplementary Table 2). For BrdU detection and 5mC/5hmC immunostainings, sections were pre-incubated in 2 N HCl for 20 min at 37 °C and neutralized in 0.1 M sodium borate (pH 8.5). Detections were performed with fluorescent secondary antibodies for 1 h at RT (Supplementary Table 3). For the study of pinwheel structures, we prepared SVZ whole-mounts using protocols established previously[8]. Briefly, the lateral walls of the lateral ventricles were dissected out and the resulting whole-mounts were fixed for 1.5 h in 4% PFA and washed overnight at 4 °C in PB. Whole-mounts for pinwheel identification were washed three times in PBS containing 0.5 % Triton X-100 for 15 min each, blocked for 2 h in 10% FBS and 2% Triton X-100 in PBS, then incubated for 48 h at 4 °C with primary antibodies for γ-tubulin to label basal bodies and β-catenin antibodies to delineate the cell membrane (Supplementary Table 2). After incubation with appropriate secondary antibodies (Supplementary Table 3), nuclei were counterstained with 1 μgml-1 of DAPI and the stained walls were mounted with Fluorsave (Calbiochem). Images were captured and analyzed with an Olympus FV10i confocal microscope (Olympus). For β-galactosidase staining brain samples were fixed with 4% PFA in 0.1 M PBS pH 7.4, 2 mM MgSO$_4$ and 5 mM EGTA for 30 minutes at 4 °C and processed for vibratome sectioning. Sections were incubated in PBS with 2 mM MgCl$_2$, 5 mM K$_3$Fe(CN)$_6$, 5 mM K$_4$Fe (CN)$_6$, 0.01% sodium deoxycholate, and 0.02% NP-40 and 1 mg ml$^{-1}$ X-Gal for 24 h at 37 °C and washed several times in PBS.

**Neurosphere cultures and knockdown experiments**. Adult 2- to 4-months-old mice were killed by cervical dislocation. To initiate each independent culture, the brains of two different animals were dissected out and the regions containing the SVZ were isolated from each hemisphere and washed in Earle's balanced salt solution (EBSS; Gibco). Tissues were transferred to EBSS containing 1.0 mg ml$^{-1}$ papain (Worthington DBA), 0.2 mg ml$^{-1}$ L-cystein (Sigma), 0.2 mg ml$^{-1}$ EDTA (Sigma), and incubated for 20 min at 37 °C. Tissue was then rinsed in EBSS, transferred to Dulbecco's modified Eagle's medium (DMEM)/F12 medium (1:1 v/v; Life Technologies) and carefully triturated with a fire-polished Pasteur pipette to a single-cell suspension. Isolated cells were collected by centrifugation, resuspended in DMEM/F12 medium containing 2 mM L-glutamine, 0.6% glucose, 9.6 g/ml putrescine, 6.3 ng/ml progesterone, 5.2 ng ml$^{-1}$ sodium selenite, 0.025 mg/ml insulin, 0.1 mg ml$^{-1}$ transferrin, 2 μg ml$^{-1}$ heparin (sodium salt, grade II; Sigma), and supplemented with 20 ng ml$^{-1}$ epidermal growth factor (EGF; Invitrogen) and 10 ng ml$^{-1}$ fibroblast growth factor (FGF; Sigma)[45,48]. For self-renewal assays, neurospheres were treated with Accutase (0.5 mM; Sigma) for 10 min, mechanically dissociated to a single-cell suspension and replated at low density (2.5 cells/μl) in growth medium (Fig. 3a). Neurospheres were allowed to develop for 6 days in a 95% air-5% CO$_2$ humidified atmosphere at 37 °C. For culture expansion, cells were plated at a relatively high density (10 cell/μl) and maintained for several passages. For cell growth assessment, a fraction of the culture at any given passage point, consisting in 250,000 viable cells, was plated and the number of cells generated was determined at the time of the next passage. To generate the accumulated cell growth curves, the ratio of cell production at each subculturing step was multiplied by the number of cells at the previous point of the curve. This procedure was repeated for each passage. For bulk differentiation assays on secondary neurospheres, 80.000 cell/cm$^2$ were seeded in Matrigel-coated coverslips and incubated 2 days in neurosphere culture medium without EGF. Medium was then changed with fresh medium without FGF but supplemented with 2% fetal bovine serum (FBS) for 5 more days (Supplementary Figure 4a). When indicated, neurosphere cultures were treated with BMP2 (R&D Systems; 20 ng/ml) or Noggin (R&D Systems; 100 ng/ml) at the time of plating. Cultures were fixed for staining at 2, 3, and 7 days of differentiation (DIV) with 4% PFA 0.1 M PBS for 15 min and

performed immunocytochemistry as described[48]. Primary and secondary antibodies and dilutions used are listed in Supplementary Table 2 and Supplementary Table 3, respectively. DAPI (1 µg/ml) was used to counterstain DNA. For cell cycle analysis $1 \times 10^6$ dissociated cells were stained using the BD Cycletest™ Plus DNA kit in accordance with the manufacture's protocol. Cell cycle phases were assessed in a FACS Verse flow cytometer (BD) and analyzed with FlowJo® software. Lentivirus containing a Snrpn-specific shRNA (TRC Clone number 0000109285, Sigma) or a non-target control (Mission SHC002, Sigma) plasmids were generated in HEK293T packing cells. Neurospheres freshly disaggregated were transduced with lentiviral particles for 6 h. Lentivirus supernatant was removed and cells allow growing for experiments.

**Luciferase reporter assays**. For luciferase assays, we electroporated $2–2.5 \times 10^6$ freshly disaggregated Tet3-Gfap$^{control}$ and Tet3-Gfap$^{cre}$ NSCs using the Amaxa NSC Nucleofector Kit, following the instructions provided by the manufacturer, with 2 µg of the BRE-tk reporter construct driving the expression of the firefly luciferase and Renilla luciferase plasmid in a 1:20 ratio. We lysed cells after 48 h and obtained cell extracts using the Dual Luciferase Reporter kit (Promega), measured luciferase activity using a VICTOR3 reader and calculated the ratio of firefly to Renilla luciferase.

**Expression studies**. RNAs were extracted with RNAeasy mini kit (Qiagen) including DNase treatment, following the manufacturer's guidelines. For quantitative PCR, 1 µg of total RNA was reverse transcribed using random primers and RevertAid H Minus First Strand cDNA Synthesis kit (Thermo Scientific), following standard procedures. Thermocycling was performed in a final volume of 10 µl, containing 1 µl of cDNA sample and the reverse transcribed RNA was amplified by PCR with appropriate Taqman probes (Supplementary Table 4). Quantitative PCR was used to measure gene expression levels normalized to Gapdh, the expression of which did not differ between the groups. qPCR reactions were performed in a Step One Plus cycler with Taqman Fast Advanced Master Mix (Applied Biosystems). SYBR green thermocycling was also performed in a final volume of 12 µl, containing 1 µl of cDNA sample, 0.2 µM of each primer (Supplementary Table 1), and SYBR® Premix ExTaq™ (Takara) according to the manufacture instructions. A standard curve made up of doubling dilutions of pooled cDNA from the samples being assessed was run on each plate, and quantification was performed relative to the standard curve.

**RNAseq**. Library preparation and high-throughput sequencing were performed by the Central Service for Experimental Research (SCSIE) at the University of Valencia. RNA-seq libraries were generated from triplicated samples per condition using the Illumina TruSeq stranded mRNA Sample Preparation Kit v2 following the manufacturer's protocol. The RNA-seq libraries were sequenced using Illumina NextSeq 500. Analysis of the RNAseq data was performed by EpiDisease S.L. The sequences were aligned to the mouse genome reference (GRCm38.p5) taken from Ensembl using the Subread package in Linux. The expression of every gene per sample was measured using a gtf annotation file (GRCm38.87) taken from Ensembl and the R package Rsubread. Data have been deposited in Gene Expression Omnibus (GEO) with the series entry is GSE127815 (https://www.ncbi.nlm.nih.gov/geo/query/acc.cgi?acc=GSE127815).

**DNA methylation and pyrosequencing**. DNA methylation level was quantified using bisulfite conversion and pyrosequencing. The DNA was bisulfited-converted using EZ DNA Methylation-Gold™ kit (Zymo research) in accordance with the manufacture's protocol. Specifically for Snrpn, bisulfite-converted DNA was amplified by PCR with specific primer pairs: Snrpn-DMR-F and Snrpn-DMR-R (Supplementary Table 1). PCRs were carried out in 20 µl, with 2U HotStar Taq polymerase (Qiagen), PCR Buffer 10×(Quiagen), 0.2 mM dNTPs, and 400 mM primers. PCR conditions were: 96 °C for 5 min, followed by 39 cycles of 94 °C for 30 s, 54 °C for 30 s, and 72 °C for 1 min. For pyrosequencing analysis, a biotin-labeled primer was used to purify the final PCR product using sepharose beads. The PCR product was bound to Streptavidin Sepharose High Performance (GE Healthcare), purified, washed with 70% ethanol, denatured with 0.2 N NaOH and washed again with 10 mM Tris-acetate. Pyrosequencing primer (400 mM) was then annealed to the purified single-stranded PCR product and pyrosequencing was performed using the PyroMark Q96MD pyrosequencing system (Qiagen) using PyroMark® reactives (Qiagen).

**Immunoblotting and ELISA**. Cells were lysed in cold RIPA buffer. Total protein concentration was determined using the BCA system (Pierce). Equal amounts (30 mg) of protein were loaded on polyacrylamide gels for SDS–polyacrylamide gel electrophoresis. Proteins were transferred to polyvinylidene difluoride (PVDF) membranes and immunoblots were carried out using primary antibodies (Supplementary Table 2) followed by incubation with appropriate secondary horse-radish peroxidase-conjugated antibodies (Supplementary Table 3) and chemoluminiscent detection (Western Lightning, PerkinElmer). All antibodies were diluted in PBS containing 5% semi-skimmed milk and 0.1% Tween-20. Proteins were revealed using Lightning® Plus ECL (Perkin Elmer) and the bands were analyzed by densitometry using ImageJ (NIH) software. Uncropped scan of

the WB gel is shown in Supplementary Figure 11a. For global determinations of 5mC and 5hmC contents, enzyme-linked immunosorbent assay (ELISA) was used. Double strand DNA was denatured at 98 °C during 10 min. DNA solutions were immobilized onto a 96-well plastic plate with Reacti-Bind DNA Coating Solution (Thermo Scientific) and incubated overnight at room temperature. Sample solutions were removed and plates were washed three times with washing buffer (0.1 M PBS containing 0.1% Tween-20) and blocked with blocking buffer (0.1 M PBS containing 5% skimmed milk and 0.1% Tween-20) at room temperature for 1 h. After washing three times with the washing buffer, primary antibodies for 5hmC or 5mC (Supplementary Table 2) were added and incubated at room temperature for 2 h. After three washes, secondary antibodies (HRP-conjugated goat anti-rabbit IgG for 5hmC or anti-mouse for 5mC, 1:2000 each, DAKO) were added and incubated at room temperature for 1 h. After three washes, One-Step Ultra TMB-ELISA (Thermo Scientific) was added to the plate to develop chemiluminescent signals, and incubated at room temperature for 30 min or until the desired color appeared. Immediately, 2 M sulfuric acid solution was added to the plate and the absorbance was measured at 450 nm using the 1420 Multilabel Counter VICTOR[3] (Perkin Elmer). The amounts of hydroxymethylated and methylated DNA were obtained by comparison to a standard curve of hydroxymethylated and methylated DNA standards (5-hmC, 5-mC, & cytosine DNA standard pack; Diagenode), respectively. The amount of total DNA examined was previously measured by the Quant-iT PicoGreen dsDNA Reagent and Kits (Invitrogen) according to the manufacturer's protocol. The percentage of 5hmC and 5mC was calculated by dividing the amount of hydroxymethylated and methylated DNA by that of total DNA, respectively.

**ChIP and 5hmeDIP**. For ChIP five 100 cm dishes with wild-type NSCs isolated from the adult SVZ were cross-linked and chromatin isolated. Chromatin was sheared to an average size of 200–500 bp using a Bioruptor sonicator (Diagenode, UCD-200). Chromatin was pre-cleared with 10 µg of non-immune rabbit IgG (Santa Cruz, cat. no. sc-2027) and 20 µl of protein G magnetic beads (Dynabeads®, cat. no. 10003D) for 3 h at 4 °C with rotation. A volume of 10 µg of TET3 antibody (Supplementary Table 2) or rabbit IgG were added and incubated overnight at 4 °C on a rotation wheel. Chromatin was precipitated with 10 µl protein G beads for 3 h at 4 °C. An aliquot of chromatin before the immunoprecipitation was used as input. Beads were then washed followed by crosslink reversal and protein digestion. Finally, DNA was purified using MiniElute PCR purification kit (Qiagen) following manufacturer's instructions. To analyze the TET3 interaction with the Snrpn promoter, ChIP enriched DNA was analyzed by qPCR using SYBR green primers (Supplementary Table 1). Pull-downs using non-immune rabbit IgG were used as control for non-specific enrichments. The comparative Ct method was used to calculate fold enrichment levels normalizing to input DNA and non-specific IgG. Uncropped scans of the PCR electrophoresis are shown in Supplementary Figure 11b.

To study TET3 binding to the paternal or maternal alleles a ChIP for TET3 was performed in NSCs derived from adult F1 mice hybrids offspring from Mus musculus domesticus (C57BL6/J) females and Mus musculus castaneus (CAST/EiJ) males (BxC hybrids NSCs), in which we identified two SNPs between the two subspecies at the Snrpn promoter (Supplementary Figure 8d). SNP1 was a "C" nucleotide in BL6 and an "A" nucleotide in Cast mice. SNP2 was a "G" nucleotide in BL6 and an "A" nucleotide in Cast mice. Genomic DNA sequences were obtained after PCR with primers within Region 2 at the Snrpn DMR (Supplementary Table 1).

To enrich DNA samples in 5hmC modification, an immunoprecipitation for 5hmC was done using 5 µg of sonicated DNA at 150–200 bp. DNA was denatured at 98 °C for 10 min and immediately cooled on ice for 10 min. Milk buffer (5% skimmed milk powder, 2 M NaCl) and 10×-IP buffer (100 mM Na-Phosphate pH 7.0, 0.5% TritonX-100) were added. DNAs were incubated with 0.5 µl of 5hmC primary antibody (Supplementary Table 2) for 2 h. In parallel, specific DNAbeads® Protein-G (Thermo Fisher) were washed with 0.1 M PBS containing 1 mg ml$^{-1}$ BSA for 2 h. Beads were then collected with a magnet, added to the DNAs and incubated overnight at 4 °C with rotation. The day after, beads were collected using a magnet and the supernatant was kept (Unbound fraction, UB). Beads were washed several times with 1x IP buffer (10 mM Na-Phosphate pH 7.0, 0.05% TritonX-100, 1 M NaCl) and both fractions were treated with 35 µg of Proteinase K (Roche) at 55 °C for 30 min. Finally, samples were purified using MiniElute PCR purification kit (Qiagen) following the manufacturer´s instructions. To calculate 5hmC enrichment at the Snrpn-DMR, qPCR was done with specific primers (Supplementary Table 1). A standard curve made up of doubling dilutions of pooled aliquots from unbound fractions was run on each plate. Fold enrichment between bound and unbound fractions was calculated estimating relative concentration with the standard curve. To ensure the specificity of 5mC and 5hmC antibodies, genomic DNAs were spiked with 0.3 ng of synthetic Arabidopsis sp. (Diagenode) sequence containing either 5-C or 5hmC and subjected to immunoprecipitation. No cross-binding was detected (Supplementary Figure 8c).

**Cloning of Tet3 variants and NSCs line construction**. Tet3 was cloned from cDNA from embryoid bodies and recombined into pDONR221 (Invitrogen). Mutations abolishing catalytic activity (HKD catalytic triad to YKA)[24,25] were introduced using the QuikChange II mutagenesis system (Agilent) using primers

Tet3-mut-F (5′-GGACTTCTGTGCCCACGCCCACAAGGACCAACATAACCT
CTACAATG-3′) and Tet3-mut-R (5′-CATTGTAGAGGTTATGTTGGTCCTT
GTGGGCGTGGGCACAGAAGTCC-3′). Variants were cloned into PB-DST-BSD
(kind gift from Jose Silva) via Gateway cloning (Invitrogen) and are expressed
constitutively from a CAG promoter. Adult NSCs lines harboring *Tet3* variants
were generated by nucleofection of $2–2.5 \times 10^6$ freshly disaggregated NSCs with 3
µg of pIG-*Tet3* or pIG-Tet3CDmut plasmids using the Amaxa NSC Nucleofector
Kit (Lonza, following the instructions provided by the manufacturer), plasmid
integration via the piggyBAC system and selection with blasticidin (10 µg/ml).
Control NSCs lines carrying a shSCRAMBLE were equally selected.

**Statistical analysis**. All statistical tests were performed using the SPSS software,
version 24.0.0 for windows, and licensed by the University of Valencia. The
significance of the differences between groups was evaluated in all using the
unpaired two-tailed Student's *t*-test or one-way ANOVA with Tukey post-hoc test
when appropriate. Treatment experiments were analyzed by paired *t*-test. When
comparisons were performed with relative values (normalized values and percen-
tages), data were normalized by using an arc-sen transformation. Values of $P < 0.05$
were considered statistically significant. Data are presented as the mean ± standard
error of the mean (s.e.m.) and the number of experiments performed with inde-
pendent cultures or animals (n) and *P*-values are indicated in the figures. Box and
whisker plots show the mean (+), median (horizontal line in box), and maximum
and minimum values.

**Reporting summary**. Further information on experimental design is available in
the Nature Research Reporting Summary linked to this article.

## Data availability
The data that support the findings of this study are available from the corresponding
author upon reasonable request. Source data for figures 1a; 2a, b, e, f, g, h; 3b, d, e, f, g, I;
4a–d, 4f–i, 5a–c, 6a, c, d, f, h; S2d, e, f, h; S3a, c, d; S4b–e; S6a, c–e, g; S7d; S8b, c and S9-c
are provided with the paper. Data that support the findings of this study have been
deposited in Gene Expression Omnibus (GEO) with the accession number GSE127815.

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

## Acknowledgements

We firstly would like to thank Dr. Isabel Fariñas for reviewing and editing the manuscript and her group for technical support and discussion of the data. We thank Elba Barberá for technical support with mice maintenance. We also thank Dr. Strogantsev for discussion of the data and Dr. Julian Peat for providing TET3 and TET3-CD mutated expression plasmids. This work was supported by grants from Ministerio de Economía y Competitividad (SAF2012-40107 and SAF2016-78845-R), Generalitat Valenciana (ACOMP2014-258), and Fundación BBVA to S.R.F. S.R.F. is a Ramón y Cajal investigator. R.M.-L. is funded by a Spanish FPI fellowship program and A.L.-U. is funded by the Generalitat Valenciana fellowship program. Work in the AFS lab was funded by grants from the MRC and Wellcome Trust.

## Author contribution

R.M.-L., A.L.-U. and S.R.F. performed most of the experiments. M.I. helped with methylation studies. C.K. contributed to *Tet3* variants overexpression experiments. S.R.F. initiated, designed, and led the study and wrote the manuscript. R.M.-L., A.L.-U, A.C.F.-S. and W.R. contributed to experimental design, data analysis, discussion, and writing of the paper.

## Additional information

**Competing interests:** The authors declare no competing interests.

**Journal Peer Review Information:** *Nature Communications* thanks the anonymous reviewer(s) for their contribution to the peer review of this work. Peer reviewer reports are available.

