## [Peer Review File · Nature Communications]

Reviewers' comments:

Reviewer #1 (Remarks to the Author):

Beginning----

In the first half, this manuscript reported an interesting finding that TET3 serves as an important regulator controlling both the pool of neural stem cells and their fate specification. In the second half, the authors showed that the TET3 (non-catalytic action) likely targets the SNRPN to repress its expression and then controls the BMP2. The findings are novel and interesting to stem cell, adult neurogenesis and neuroscience communities. Most of conclusions have been well supported by current data collections if having been precisely described. However, the second half on SNRPN and BMP2 is not fully convincing yet, still can be co-incident. More importantly, this manuscript as its current format, can be greatly improved, and more importantly some descriptions are not precise. Although readable, the manuscript will benefit from being extensively edited by an experienced expert in the adult SVZ field.

Major concerns

1) Some conclusions are not well supported by the data (i.e., figure 3h, i), or assumed from the data (i.e., figure 1b referring to mature neurons). Although most won't affect the conclusions, this issue across the the whole manuscript made it hard to appreciate the conclusions (some have been detailed in the minor comments for individual figures).

2) Many analyses in details were not well addressed. For example, in figure 2, most normalization was not clearly justified or described (the Figure 2 was exemplified in the minor comments)

3) The TET3-SNRPN (BMP2) conclusion needs further evidence to be strengthened. The current data collection (Figure 6 e and f) is insufficient to make a strong conclusion as the authors claimed (detailed in the minor comments).

Minor concerns

In the Figure 1, the authors performed a cluster of staining and mRNA tests showing that the Tet3 is expressed in adult neural stem cells. The data is enriched and seems convincing, however remains confusing from what the authors presented. a), does the SVZ represent a mixture of cells? Where were these NSCs from? For the 2-7 DIV, were the NSCs used a mixture of different types of cells. The authors should clarify these. Since there are error bars, a statistical comparison should be performed. b), the conclusion of TET3 expression in mature neurons is short of evidence without any mature neuronal marker staining. A staining with SOX2 should be added to confirm the observation. DCX+ cells in this image seem to be neuroblasts, but due to DCX staining immature neurons if the authors want to make this conclusion a more specific marker will be required. d), in a) there is no evidence showing that TET3 is in astrocytes. Also, the conclusion is somewhat odd, but I assume that the authors wanted to conclude that the TET3 is expressed in neural stem cells of SVZ, and GFAP+ NSCs and its lineage (at least from the images).

In the figure 2, the authors found that the deletion of TET3 using the GFAP-cre line (constitutive expression of Cre in GFAP+ cells including astrocytes) decreased the NSC pool size. But likely with biased generation of GFAP+ cells. The b), c), e), f) (pinwheel analysis), g) and h) showed the decreased pool or resulted from the decreased pool. The d) and part f) (Ast) showed the biased fate selection. The conclusion from the supplementary figure 1 is not convincing since the GFAP-Cre line is not a NSC-specific mouse line. d), what did the authors normalize to, all the SVZ cells? Why not BrdU+ cells as described in b). What is the portion of DCX+? Or SOX2?

In the figure 3, this cluster of experiments showed that the Tet3 depletion led to biased GFAP+ fate. The data seems sufficient, it will be necessary to present the S100beta data together with either Nestin or another immature neuronal marker to make this conclusion.

In the figure 4-6, the elevated expression of Snrpn is exciting. In the figure 5, e) and f), it will be more convincing if with a Noggin in control. The comparison between snSCRAM and shSNRPN should be performed. Further, the experiments in Figure 6e and f will be necessary to be expanded. For example, will expression of shSNRPN abolish the effect of Tet3 expression, etc. This sort of tests will advance our understanding whether these changes are indeed correlated.

In addition, in most figures, the statistic analyses should be revisited.

---Ending

Reviewer #2 (Remarks to the Author):

Reviewer summary:

In this manuscript, Montalbán-Loro et al. explore the role of Ten-eleven translocation methylcytosine dioxygenase 3 (TET3) in neural stem cells within the subventricular zone (SVZ) of the mouse brain. To aid in their studies, they generated TET3-deficient NSCs in the SVZ of mice via a cross between a floxed Tet3 mouse and a mouse driving Cre-recombinase in Gfap-expressing cells. They provide substantial evidence (Fig.2, Fig. s1) that their model system is working and that TET3 has indeed been removed. From here, they demonstrate that there is a significant reduction in NSC self-renewal activity and a skewing of the population such that a higher proportion of cells have adopted a differentiated, astrocytic fate (S100β) in TET3-deficient SVZs (in vivo) and neurospheres (in vitro) compared to controls.

Through a RNA-seq experiment, the authors find that the imprinted gene Small nuclear ribonucleoprotein-associated polypeptide N (Snrpn) is significantly upregulated in Tet3-Gfapcre (KO) NSCs compared to controls. To test whether increased Snrpn expression was responsible for the TET3-mediated NSC defects, the authors knocked down Snrpn and revealed that many of the defects observed in TET3-deficient NSCs were reverted back to control levels. These data suggest that TET3 might normally act to regulate Snrpn expression in NSCs, and that its absence leads to increased expression Snrpn and disrupted NSC function.

To understand how Snrpn might disrupt NSC function, the authors turned their attention to the Bone Morphogenic Protein (BMP) signaling cascade because of previous reports indicating that the pathway promotes gliogenesis and therefore, might explain the increased astrocyte numbers observed in Tet3-Gfapcre KO cells. Using a number of different experimental techniques, the authors demonstrate there is in fact an increase in BMP signaling in Tet3-Gfapcre KO cells and that KD of Snrpn can block this increase. Thus, taken together, these data suggest that loss of TET3 leads to defects in self renewal and increased astrocyte differentiation in NSCs through an increase in Snrpn expression and an upregulation in BMP signaling.

Finally, the authors explore how TET3 might be affecting the expression of Snrpn, by measuring 5mC and 5hmC levels in NSCs and performing ChIP experiments to examine if the methylcytosine dioxygenase binds to the Snrpn promoter region. Although the authors observed no significant differences in 5mC and 5hmC that could explain how TET3 influences Snrpn expression, they did find that it binds directly to the Snrpn promoter. Accordingly, using TET3 constructs, the researchers go on to show that TET3 negatively regulates Snrpn expression in a specific manner and that it appears to

function independent of its catalytic activity (i.e. 5mC to 5hmC conversion).

Reviewer suggestions for the authors:

1) In Figures 1b-d the authors use the anti-Tet3 antibody ABE290 for IHC and ICC images of TET3 protein localization both in the SVZ and in neurosphere cultures. It is somewhat concerning that in some of the images (in particular 1b) that the immunostaining for TET3 appears to be rather nonspecific as most of it is not present in the DAPI stained nuclei. It would be helpful if the authors could provide some supplemental data either through WBs in SVZ or perhaps through overexpression/or KD of Tet3 in cell culture (especially given that they have the constructs) as a validation of this antibody's specificity.

2) This a fairly minor point, but in supplement figure S1d, the authors show images of TET3 IHC signal reduction in neurospheres from Tet3-Gfapcre mice. However, there is no comparison to a control neurosphere from which to compare TET3 signal. If the authors are going to show an image of TET3 reduction by IHC in the Tet3-Gfapcre neurospheres there must be a control image to compare it to. Alternatively, given that the WB in S1c already demonstrates reduced TET3 levels, 1d could be excluded from the figure due to redundancy.

3) In Figure 2b, the authors show there is a reduction in GFAP+ BrdU+ and GFAP/Ki67+ BrdU+ cells in Tet3-Gfapcre animals. As the authors have consistently provided IHC or ICC images that accompany their other quantitative histograms, I would like to see at least one image from both groups showing BrdU staining. These images could either fit in next to Figure 2b or could be placed in supplemental materials.

Figure 2h also involves BrdU staining in glomeruli of the OB. Same recommendation as above for Fig. 2b.

4) Related to Figure 2e, f, the authors write, "Consistently, a decrease in the number of GFAP+ and γ -tubulin+ NSCs in the pinwheels contacting the lateral ventricle was found, together with an increase in the terminally differentiated astrocytes in the Tet3-Gfapcre SVZ ventricle wall (Fig. 2e, f)." Given that this journal is meant for a broad audience, it would be beneficial to either put something in the introduction about the pin wheel structures or use a sentence to briefly explain what they are so that researchers not directly in the SVC NSC field understand the significance of this finding. Also, in the methods section there is no mention of how these structures are ID'ed or quantified. A reference may be sufficient, along with the statement "as previously described".

5) Related to Figure 3, the authors state, "This was confirmed by immunostaining for Caspase 3+ cells in neurosphere cultures from the two genotypes (relative to DAPI+ cells: 3.8 ± 0.5 % in wild-type cultures and 4.2 ± 0.3 in Tet3-Gfapcre). However, I couldn't locate a figure that accompanied this statement. The authors should either insert "data not shown" at the end of the sentence or provide an image somewhere demonstrating the Caspase-3 staining.

6). In the sentence, containing "whereas no change in the neuronal DIII-tubulin gene (Tubb3) was found (Fig. S3b)", S3b should be changed to S2b.

7) In Figure S4 the authors provide a heat map from their RNA-seq experiment comparing gene expression between Tet3-Gfapcontrol and Tet3-Gfapcre samples. There are several issues with the authors presentation of this data that needs to be further clarified.

A) There is no label or mention in the figure legend about what the colors represent, (i.e. FPKM, RPKM, log fold change, etc...). Likewise, it appears that there are some genes present that show no changes in color between groups, suggesting no difference, yet are still in the top 20% of genes

differentially expressed. To summarize this point, more information is needed here so that the readers can correctly interpret the heat map.

B) They show only 1/5 of the statistically significant genes mentioned in the text of their results section. This data should be provided in a supplemental table.

C) A table of the FC or FPMK for the imprinted genes should also be provided in the supplemental section.

8) In the results section titled "TET3 regulates the maintenance and differentiation of NSCs through repression of *Snrpn*". The authors go from talking about the RNAseq data straight to imprinted genes without a brief explanation (1 or 2 sentences even) of why. It is not clear to the reader why the study goes in this direction, nor is this emphasized in the introduction. The authors do however, refer to papers demonstrating that imprinting is important in stem cells (ref. 55) and would be a nice tie-in to set up this transition.

9) In figure 4D, the authors demonstrate the effectiveness of their LV-shSNRPN tool to KD the expression of *Snrpn* mRNA levels in both Tet3-GFAPcontrol and Tet3-GFAPcre, but it appears that the shRNA does not significantly KD *Snrpn* in the control group, but only lowers it in the experimental group. How is this the case? Perhaps it does, but the authors didn't perform the comparison. It should KD the expression no matter what sample it is. I would be nice if the authors could at least provide the reviewers evidence demonstrating that it works in both groups, but of course the effect is more dramatic in the Tet3-GFAPcre samples.

10) The authors provide ample evidence in this manuscript that supports the idea that loss of Tet3 in SVZ NSCs disrupts self-renewal and skews that population towards increased levels of differentiated astrocytic cells, positive for S100 β . Moreover, their data strongly suggest that this disruption appears to result from an apparent non-catalytic role for TET3 in regulating the expression of the imprinted gene *Snrpn*. As a consequence of increased *Snrpn*, BMP signaling is upregulated in SVC NSCs, which is sufficient to drive a larger than normal percentage of these cells to adopt an astrocytic fate. Given that the authors examine this phenomenon on so many levels it would be extremely helpful to the reader to provide an illustration of their TET3-*Snrpn*-BMP model and how it regulates self-renewal and astrocytic fate in SVC NSCs (pretty much a visual abstract). Ideally, this would be found at the end of the results section, if space allows.

We thank the reviewer for her/his comments and for the suggestions which have significantly improved our work.

Answers to reviewers' comments: Each concern/comment made the reviewers (highlighted in bold) is listed and followed by our response and the corresponding addition/alteration we have made to generate a revised version of the manuscript.

Reviewer #1

Beginning----

In the first half, this manuscript reported an interesting finding that TET3 serves as an important regulator controlling both the pool of neural stem cells and their fate specification. In the second half, the authors showed that the TET3 (non-catalytic action) likely targets the SNRPN to repress its expression and then controls the BMP2. The findings are novel and interesting to stem cell, adult neurogenesis and neuroscience communities. Most of conclusions have been well supported by current data collections if having been precisely described. However, the second half on SNRPN and BMP2 is not fully convincing yet, still can be co-incident.

More importantly, this manuscript as its current format, can be greatly improved, and more importantly some descriptions are not precise. Although readable, the manuscript will benefit from being extensively edited by an experienced expert in the adult SVZ field.

Major concerns

1) Some conclusions are not well supported by the data (i.e., figure 3h, i), or assumed from the data (i.e., figure 1b referring to mature neurons). Although most won't affect the conclusions, this issue across the whole manuscript made it hard to appreciate the conclusions (some have been detailed in the minor comments for individual figures). 2) Many analyses in details were not well addressed. For example, in figure 2, most normalization was not clearly justified or described (the Figure 2 was exemplified in the minor comments). 3) The TET3-SNRPN (BMP2) conclusion needs further evidence to be strengthened. The current data collection (Figure 6 e and f) is insufficient to make a strong conclusion as the authors claimed (detailed in the minor comments).

We fully agree with the reviewer in that it is important to analyze more thoroughly the relation of SNRPN with BMP2 and with the phenotype observed in *Tet3* deficient NSCs. To strength the correlation between TET3, SNRPN and the downstream regulation of BMP2 signalling to control the astroglial terminal differentiation of NSCs, we have performed additional experiments that are explained in detail in the minor comments.

In an effort to provide a more precise description of the astroglial phenotype found in *Tet3* deficient SVZ, we have asked Dr. Isabel Fariñas, full professor at the University of Valencia and EMBO member with a wide expertise in the *in vivo* and *in vitro* regulation of SVZ adult neural stem cells (ORCID: 0000-0003-2903-4960), for an extensive reading and edition of the manuscript. She has kindly accepted to revise the text and the data provided and has suggested several changes that are currently incorporated to the new version of the manuscript.

Minor concerns

In the Figure 1, the authors performed a cluster of staining and mRNA tests showing that the *Tet3* is expressed in adult neural stem cells. The data is enriched and seems convincing, however remains confusing from what the authors presented.

a), does the SVZ represent a mixture of cells? Where were these NSCs from? For the 2-7 DIV, were the NSCs used a mixture of different types of cells. The authors should clarify these. Since there are error bars, a statistical comparison should be performed.

The SVZ data is generated from dissected tissue of adult mouse brain mice, thus includes NSCs, neural progenitors, neuroblasts, ependymal cells and the mixture of cells within the niche, including blood vessels. NSCs are isolated from the SVZ neurogenic niche and expanded for several passages in a selection medium containing basic fibroblast growth factor (FGF) and epidermal growth factor (EGF) as mitogens to produce multipotent clonal aggregates as we have previously described (Ferrón et al., 2007; Belenguer et al., 2016). Mitogen withdrawal in the culture medium allows the study of the differentiation potential of NSCs, thus 2-7 DIV conditions represent different steps of differentiation from NSCs cultures and include neurons, astrocytes and oligodendrocytes (Belenguer et al., 2016). We have now clarified this in Methods in the new version of the manuscript.

As the reviewer indicates we have now performed a One-way ANOVA and Tukey post-test to determine the differences between samples in **Figure 1a**. We have included the p-values and significance in the graphs and in the figure legend.

b), the conclusion of TET3 expression in mature neurons is short of evidence without any mature neuronal marker staining. A staining with SOX2 should be added to confirm the observation. DCX+ cells in this image seem to be neuroblasts, but due to DCX staining immature neurons if the authors want to make this conclusion a more specific marker will be required. c), in a) there is no evidence showing that TET3 is in astrocytes. Also, the conclusion is somewhat odd, but I assume that the authors wanted to conclude that the TET3 is expressed in neural stem cells of SVZ, and GFAP+ NSCs and its lineage (at least from the images).

We agree with the reviewer that this is confusing.

As the reviewer mentions, our intention is to describe the level of expression of *Tet3* in the neural stem cell population, which present astrocytic characteristics, such as the glial fibrillary acidic protein (GFAP) and that proliferate and self-renew *in vivo* and *in vitro*. Therefore, to better clarify this point we have performed the immunostaining for TET3 together with GFAP and the neural stem cell marker SOX2. We have also performed the immunodetection for TET3 in combination with the neuron-specific cytoskeletal protein MAP2 expressed in mature neurons. Images for the co-localization of TET3 in the GFAP/SOX2+ NSC population are included in new **Figure 1b**. Images of the co-localization of TET3 and MAP2 in mature neurons in the striatal parenchyma are also included in new **Figure S1b**.

Terminally differentiated astrocytes express GFAP and S100 β protein that is largely absent from multipotent GFAP+ stem cells and correlates with loss of neurosphere forming potential (Raponi et al., 2007). To determine, the levels of expression of *Tet3* also in differentiated astrocytes, we have isolated primary astrocytes from the adult SVZ and cultured them *in vitro* in differentiation conditions (Ferrón et al., 2011). The absence of mitogens and the presence of fetal bovine serum in the medium induce terminal differentiation of these cells and promote the expression of GFAP and S100 β markers after 7 days of culture (Raponi et al., 2007; Ferrón et al., 2011). We have performed an mRNA expression analysis for the *Tet* genes in these cells and found that only *Tet2* is present, whereas *Tet1* and *Tet3* are almost absent terminally differentiated astrocytes. These data have been added to new **Figure 1a**.

These new studies confirm that TET3 is present in the GFAP/SOX2+ NSCs in the SVZ but not in the DCX+ neuroblast (**Figure S1a**) or S100 β + mature astrocytes populations. We also confirm the presence of the enzyme in some mature neurons in the striatal parenchyma.

In the figure 2, the authors found that the deletion of TET3 using the GFAP-cre line (constitutive expression of Cre in GFAP+ cells including astrocytes) decreased the NSC pool size. But likely with biased generation of GFAP+ cells. The b), c), e), f) (pinwheel analysis), g) and h) showed the decreased pool or resulted from the decreased pool. The d) and part f) (Ast) showed the biased fate selection. The conclusion from the supplementary figure 1 is not convincing since the GFAP-Cre line is not a NSC-specific mouse line.

We understand reviewer's concern about the specificity of the GFAP-cre mouse line.

As we mention in the manuscript, mice expressing cre-recombinase directed by the mouse glial fibrillary acidic protein (*Gfap*) promoter are from Jackson Lab (B6.Cg-Tg(*Gfap-cre*)73.12Mvs/J; Stock No. 012886) (Garcia et al., 2004). The company reports that in contrast to other GFAP-cre lines (for example 77.6 line Stock 024098), these mice have cre-recombinase activity in essentially all adult neural stem cells (and their progeny) from the dentate gyrus and subventricular zone. Moreover, X-gal histochemistry in the adult brain of ROSA26R; *Gfap-cre*⁺⁰ mice did not show positive staining in the majority of GFAP+ cells within the striatal parenchyma, indicating that recombination did not occur in mature astrocytes. We have performed an immunostaining for GFAP and β -galactosidase in the SVZ and confirmed the specific recombination mainly in GFAP+ adult stem cells (new **Figure S2c**).

To further address reviewer's concern, we have isolated primary astrocytes from the adult SVZ of two months-old *Tet3-Gfap*^{cre} and *Tet3-Gfap*^{control} mice, as we previously described (Ferrón et al., 2011), and determined the levels of expression of *Tet3* in terminally differentiated astrocytes after recombination (Ferrón et al., 2011). This analysis confirmed the low levels of expression of the gene in wild-type mature astrocytes compared to NSC. Moreover, although *Tet3* was slightly decreased in *Tet3-Gfap*^{cre} compared to *Tet3-Gfap*^{control} astrocytes, this might be due to the neural stem cell origin of these primary astrocytes, but not to a specific recombination in differentiated astrocytes. These data have been added to new **Figure 2a**.

d), what did the authors normalize to, all the SVZ cells? Why not BrdU+ cells as described in b).

Figure 2b shows the total number of BrdU-LRCs that are GFAP+ (left panel) and the percentage of GFAP/LRC+ that express the cell cycle marker Ki67+. **Figures 2e** and **2g** show the percentage of cells and are normalized to the total DAPI. In **Figures 2f** and **2h** the data are normalized to the area analyzed. This has now been indicated in the graphs.

What is the portion of DCX+? Or SOX2?

In the previous manuscript we included the quantification of the levels of DCX staining of confocal reconstructions of whole-mounts of the rostral migratory stream of mice from both genotypes. We have now determined the percentage of DCX+ cells normalized to total number of DAPI in *Tet3-Gfap^{control}* and *Tet3-Gfap^{cre}* SVZ. This has been included as new **Figure 2g** together with confocal images of the DCX immunostaining.

In the figure 3, this cluster of experiments showed that the Tet3 depletion led to biased GFAP+ fate. The data seems sufficient, it will be necessary to present the S100beta data together with either Nestin or another immature neuronal marker to make this conclusion.

As the reviewer suggests we have included images and the quantification for the progenitor marker Nestin. This data are now in **Figure 3 h** and **i**.

In the figure 4-6, the elevated expression of *Snrpn* is exciting.

In the figure 5, e) and f), it will be more convincing if with a Noggin in control.

Old **Figures 5a, b** and **c** are now **Figure 4 h, i** and **j** respectively, and we have included data with Noggin treatment in control cells.

The comparison between snSCRAM and shSNRPN should be performed.

We have now provided the comparisons requested.

Further, the experiments in Figure 6e and f will be necessary to be expanded. For example, will expression of shSNRPN abolish the effect of Tet3 expression, etc. This sort of tests will advance our understanding whether these changes are indeed correlated.

We have previously showed that the use of a shSNRPN to silence *Snrpn* caused a rescue of the premature astrocytic differentiation phenotype in *Tet3-Gfap^{cre}* neurosphere cultures through the recovery of *Bmp2* to control levels (old **Figures 4** and **5**). We have also showed that the overexpression of *Tet3* or of a catalytically inactive variant of *Tet3* (*Tet3* CDmut) in *Tet3-Gfap^{cre}* NSCs, caused a significant decrease in *Snrpn* expression which led to an increase in the number of spheres formed in self-renewal conditions (old **Figures 6e** and **f**).

To determine whether the overexpression of the enzyme had an effect also in the premature astrocytic differentiation of *Tet3* deficient NSCs, we have overexpressed *Tet3* and *Tet3* with the catalytic domain mutated (*Tet3* CDmut) in *Tet3-Gfap^{cre}* NSCs. We first confirmed that overexpression of *Tet3* or *Tet3* CDmut, causes a decrease of the expression of the *Snrpn* gene in *Tet3-Gfap^{cre}* neurosphere cultures. Consistent to previous data, this led to an increase in the number of spheres formed and restored the percentage of S100β+ cells in *Tet3-Gfap^{cre}* differentiated cultures to control levels, confirming that *Tet3* regulates the differentiation capacity of NSCs and that is not dependent on its catalytic function. This new experiments have been added as **Figures 6c-f**.

Importantly, the decrease of *Snrpn* after overexpression of *Tet3* or its catalytically inactive variant led to a decrease in the levels of pSMAD1/5 and in the basal activity of the BRE reporter in *Tet3-Gfap^{cre}*, linking *Snrpn* regulation to the BMP2 signaling. Therefore, our data demonstrate that TET3 binds to the *Snrpn* promoter at the paternal allele to contribute to transcriptional repression of the paternal allele in adult NSCs, independently of its catalytic function. Our data also involves BMP2 as a responsible for the astrocytic terminal differentiation mediated by SNRPN in *Tet3* deficient NSCs. Similar experiments in *Tet3-Gfap^{control}*, did not show any change in the *Snrpn* or *Bmp2* levels, or self-renewal and cell differentiation, suggesting that higher levels of TET3 in wild-type NSCs do not alter their response. These new data have been added as **Figures 6g, h** and **S9**.

In addition, in most figures, the statistic analyses should be revisited.

Statistics have been revisited and we have indicated the tests used in each Figure.

---Ending

Reviewer #2:

Reviewer summary:

In this manuscript, Montalbán-Loro et al. explore the role of Ten-eleven translocation methylcytosine dioxygenase 3 (TET3) in neural stem cells within the subventricular zone (SVZ) of the mouse brain. To aid in their studies, they generated TET3-deficient NSCs in the SVZ of mice via a cross between a floxed Tet3 mouse and a mouse driving Cre-recombinase in Gfap-expressing cells. They provide substantial evidence (Fig.2, Fig. s1) that their model system is working and that TET3 has indeed been removed. From here, they demonstrate that there is a significant reduction in NSC self-renewal activity and a skewing of the population such that a higher proportion of cells have adopted a differentiated, astrocytic fate (S100 β) in TET3-deficient SVZs (in vivo) and neurospheres (in vitro) compared to controls.

Through a RNA-seq experiment, the authors find that the imprinted gene Small nuclear ribonucleoprotein-associated polypeptide N (Snrpn) is significantly upregulated in Tet3-Gfapcre (KO) NSCs compared to controls. To test whether increased Snrpn expression was responsible for the TET3-mediated NSC defects, the authors knocked down Snrpn and revealed that many of the defects observed in TET3-deficient NSCs were reverted back to control levels. These data suggest that TET3 might normally act to regulate Snrpn expression in NSCs, and that its absence leads to increased expression Snrpn and disrupted NSC function.

To understand how Snrpn might disrupt NSC function, the authors turned their attention to the Bone Morphogenic Protein (BMP) signaling cascade because of previous reports indicating that the pathway promotes gliogenesis and therefore, might explain the increased astrocyte numbers observed in Tet3-Gfapcre KO cells. Using a number of different experimental techniques, the authors demonstrate there is in fact an increase in BMP signaling in Tet3-Gfapcre KO cells and that KD of Snrpn can block this increase. Thus, taken together, these data suggest that loss of TET3 leads to defects in self renewal and increased astrocyte differentiation in NSCs through an increase in Snrpn expression and an upregulation in BMP signaling.

Finally, the authors explore how TET3 might be affecting the expression of Snrpn, by measuring 5mC and 5hmC levels in NSCs and performing ChIP experiments to examine if the methylcytosine dioxygenase binds to the Snrpn promoter region. Although the authors observed no significant differences in 5mC and 5hmC that could explain how TET3 influences Snrpn expression, they did find that it binds directly to the Snrpn promoter. Accordingly, using TET3 constructs, the researchers go on to show that TET3 negatively regulates Snrpn expression in a specific manner and that it appears to function independent of its catalytic activity (i.e. 5mC to 5hmC conversion).

Reviewer suggestions for the authors:

1) In Figures 1b-d the authors use the anti-Tet3 antibody ABE290 for IHC and ICC images of TET3 protein localization both in the SVZ and in neurosphere cultures. It is somewhat concerning that in some of the images (in particular 1b) that the immunostaining for TET3 appears to be rather nonspecific as most of it is not present in the DAPI stained nuclei. It would be helpful if the authors could provide some supplemental data either through WBs in SVZ or perhaps through overexpression/or KD of Tet3 in cell culture (especially given that they have the constructs) as a validation of this antibody's specificity.

We understand reviewer's concern about the specificity of the TET3 antibody. We have now performed an IHC with a new antibody anti-TET3 from Abnova (Cat.#: PAB25635). We provide new images for the detection of TET3 in GFAP/SOX2+ stem cell population *in vivo* in which most of the previously observed non-specific cytoplasmic background is disappeared (new **Figure 1b**). We also provide images for the detection of TET3 and DCX in the neuroblast population and in MAP2+ mature neurons within the parenchyma (new **Figures S1a** and **S1b**).

We have also performed an ICC using the same antibody *in vitro*, neurosphere of *Tet3-Gfap^{control}* and *Tet3-Gfap^{cre}* cultures. Images for this staining are included in new **Figure S2g**.

As the reviewer suggested we have also used *Tet3-Gfap^{cre}* cultures overexpressing *Tet3* to perform the ICC and validate the antibody specificity. Images of these ICC have been added as new **Figures 6e** and **S9d**.

2) This is a fairly minor point, but in supplement figure S1d, the authors show images of TET3 IHC signal reduction in neurospheres from *Tet3-Gfap^{cre}* mice. However, there is no comparison to a control neurosphere from which to compare TET3 signal. If the authors are going to show an image of TET3 reduction by IHC in the *Tet3-Gfap^{cre}* neurospheres there must be a control image to compare it to. Alternatively, given that the WB in S1c already demonstrates reduced TET3 levels, 1d could be excluded from the figure due to redundancy.

As we mentioned in the previous point, we have performed an ICC using a new anti-TET3 antibody in *Tet3-Gfap^{control}* and *Tet3-Gfap^{cre}* cultures. Images of these ICCs have been added as new **Figure S2g**.

3) In **Figure 2b**, the authors show there is a reduction in GFAP+ BrdU+ and GFAP/Ki67+ BrdU+ cells in *Tet3-Gfap^{cre}* animals. As the authors have consistently provided IHC or ICC images that accompany their other quantitative histograms, I would like to see at least one image from both groups showing BrdU staining. These images could either fit in next to **Figure 2b** or could be placed in supplemental materials. **Figure 2h** also involves BrdU staining in glomeruli of the OB. Same recommendation as above for **Fig. 2b**.

As the reviewer suggested we have now included additional images of the IHC from the different markers. In new **Figure 2c** we have included a representative image for the BrdU detection in *Tet3-Gfap^{control}* and *Tet3-Gfap^{cre}* SVZ. Additionally we have incorporated higher magnification images for the BrdU labelling combined with the GFAP marker (new **Figure 2d**) and for GFAP combined with SOX2 (new **Figure S3a**) within the SVZ of both genotypes. Finally we have included images for the BrdU staining in the OB of *Tet3-Gfap^{control}* and *Tet3-Gfap^{cre}* mice (new **Figure 2h**).

4) Related to **Figure 2e, f**, the authors write, “Consistently, a decrease in the number of GFAP+ and γ -tubulin+ NSCs in the pinwheels contacting the lateral ventricle was found, together with an increase in the terminally differentiated astrocytes in the *Tet3-Gfap^{cre}* SVZ ventricle wall (Fig. 2e, f).” Given that this journal is meant for a broad audience, it would be beneficial to either put something in the introduction about the pin wheel structures or use a sentence to briefly explain what they are so that researchers not directly in the SVC NSC field understand the significance of this finding. Also, in the methods section there is no mention of how these structures are ID'ed or quantified. A reference may be sufficient, along with the statement “as previously described”.

We agree with the reviewer that this is not clearly described in the manuscript.

We have now included a paragraph (below) in the introduction describing how the walls of the lateral ventricles are organized in pinwheel structures with type B cells surrounded by a rosette of epithelial ependymal cells (Mirzadeh et

“In the SVZ in particular, NSCs also known as type B-cells have a long basal process ending on blood vessels{Tavazoie, 2008 #124} and extend an apical process ending in a primary cilium that protrudes into the ventricle{Doetsch, 1997 #116}. The walls of the lateral ventricles thus show a typical organization where the apical process of type B cells are surrounded by a rosette of epithelial ependymal cells forming structures known as pinwheels{Mirzadeh, 2010 #56}.”

To quantify the number of pinwheels images of the SVZ wall were captured and analyzed with an Olympus FV10i confocal microscope (Olympus). The number of pinwheels per area was estimated considering a rosette of several ependymal cells surrounding type B cilia. Moreover, monociliated GFAP+ cells with their cell bodies completely intercalated within the ependymal layer correspond to terminally differentiated astrocytes, which are reported to increase with aging (Luo et al 2006). These technical details have been added to the manuscript. We have also included a better description (below) of how we identify these structures in the methods section

*“For the study of pinwheel structures, we prepared SVZ whole-mounts using protocols established previously⁹. Briefly, the lateral walls of the lateral ventricles were dissected out and the resulting whole-mounts were fixed for 1.5 h in 4% PFA and washed overnight at 4 °C in PB. Whole-mounts were washed three times in PBS containing 0.5 % Triton X-100 for 15 min each, blocked for 2 h in 10% FBS and 2% Triton X-100 in PBS, then incubated for 48 h at 4 °C with primary antibodies for γ -tubulin to label basal bodies and β -catenin antibodies to delineate the cell membrane (see **Table S1a**). After incubation with appropriate secondary antibodies, the stained walls were mounted with Fluorsave (Calbiochem). Nuclei were counterstained with 1 μ g/ml of DAPI.”*

5) Related to Figure 3, the authors state, “This was confirmed by immunostaining for Caspase 3+ cells in neurosphere cultures from the two genotypes (relative to DAPI+ cells: 3.8 ± 0.5 % in wild-type cultures and 4.2 ± 0.3 in Tet3-Gfapcre). However, I couldn’t locate a figure that accompanied this statement. The authors should either insert “data not shown” at the end of the sentence or provide an image somewhere demonstrating the Caspase-3 staining.

New images for the immunodetection and quantification of activated Caspase-3 in *Tet3-Gfap^{control}* and *Tet3-Gfap^{cre}* cultures have been added as new Figure S3d.

6). In the sentence, containing “whereas no change in the neuronal DIII-tubulin gene (*Tubb3*) was found (Fig. S3b)”, S3b should be changed to S2b.

This has been changed and is in new Figure S4b.

7) In Figure S4 the authors provide a heat map from their RNA-seq experiment comparing gene expression between Tet3-Gfapcontrol and Tet3-Gfapcre samples. There are several issues with the authors presentation of this data that needs to be further clarified.

A) There is no label or mention in the figure legend about what the colors represent, (i.e. FPKM, RPKM, log fold change, etc...). Likewise, it appears that there are some genes present that show no changes in color between groups, suggesting no difference, yet are still in the top 20% of genes differentially expressed. To summarize this point, more information is needed here so that the readers can correctly interpret the heat map.

We apologise for the lack of clarity in the figure originally provided.

We have now indicated in the figure and its legend that the colours represent the normalized reads counts value for each sample, which indicates the abundance of a specific gene. Yellow line in the colour legend indicates the number of genes for each value, showing that the majority of the genes increase their expression in *Tet3-Gfap^{cre}* cultures relative to controls. This has been indicated in the figure and figure legend. We have also modified the colours of the heat map to avoid colour saturation and to show more clearly the differences between groups (new Figure S5a).

B) They show only 1/5 of the statistically significant genes mentioned in the text of their results section. This data should be provided in a supplemental table. C) A table of the FC or FPMK for the imprinted genes should also be provided in the supplemental section.

As the reviewer suggested we have now included a supplemental table with the list of significantly upregulated and downregulated genes obtained from the RNAseq analysis after depletion of *Tet3* in adult NSCs (Table S1e and f respectively). We have also provided the list of imprinted genes highlighting in bold the upregulated (blue) and downregulated (pink) genes (Table S1e). LogFC and FDR are indicated. We considered that a gene was significantly changed when the FDR for that gene was less than 0.05 and is indicated in the Table legend.

8) In the results section titled “TET3 regulates the maintenance and differentiation of NSCs through repression of *Snrpn*”. The authors go from talking about the RNAseq data straight to imprinted genes without a brief explanation (1 or 2 sentences even) of why. It is not clear to the reader why the study goes in this direction, nor is this emphasized in the introduction. The authors do however, refer to papers demonstrating that imprinting is important in stem cells (ref. 55) and would be a nice tie-in to set up this transition.

We fully agree with the reviewer in that it is important to explain why we focus on the regulation of imprinted genes expression in the RNAseq data. Thus, we have rephrased this results section and included, in the introduction, a paragraph (below) highlighting the relevance of the regulation of genomic imprinting in NSC function.

“Most mammalian genes are expressed from both maternally and paternally inherited chromosomal homologues. In contrast, imprinted genes are expressed from one parental copy only and the gene copy inherited from the other progenitor remains repressed from the zygote{Ferguson-Smith, 2011 #126}. Imprinting is regulated by epigenetic mechanisms, in particular DNA methylation imprints, that establish and maintain parental identity{Edwards, 2007 #127}. Their monoallelic expression makes these loci very vulnerable as mutation or deregulation of the sole expressed allele can compromise expression and lead to severe developmental defects{Cleaton, 2014 #129;Wilkinson, 2007 #128}. Interestingly, recent evidences have suggested that selective absence of imprinting can occur in particular lineages to modulate the dosage of imprinted genes for cell-specific functions{Ferron, 2011

#55;Ferron, 2015 #119}. These changes appear to be context-dependent and may be important for cell plasticity and normal development and tissue regeneration{Kar, 2014 #125}. Indeed, we have previously described that in the SVZ the paternally-expressed gene *Delta-like homologue 1 (Dlk1)*, an atypical Notch ligand, plays a relevant function in postnatal neurogenesis. *Dlk1* is canonically imprinted elsewhere in the brain, however, it shows a selective absence of imprinting in NSCs with biallelic expression being required for stem cell maintenance and, ultimately, neurogenesis to the OB{Ferron, 2011 #55}. Another imprinted gene, the *Insulin-like growth factor2 (Igf2)*, which is canonically expressed from the paternally-inherited allele, is biallelically expressed in the choroid plexus and secreted into the cerebrospinal fluid to regulate NSC proliferation{Ferron, 2015 #119;Lehtinen, 2011 #120}, whereas it is imprinted in the hippocampus NSCs acting as an autocrine factor{Ferron, 2015 #119}. Therefore, determining how imprinted genes operate in concert with signaling cues, as well as discovering the factors that modify methylation at imprinted regions in adult NSCs of different neurogenic niches, will lead to a better understanding of adult neurogenesis.“

9) In figure 4D, the authors demonstrate the effectiveness of their LV-shSNRPN tool to KD the expression of *Snrpn* mRNA levels in both Tet3-GFAPcontrol and Tet3-GFAPcre, but it appears that the shRNA does not significantly KD *Snrpn* in the control group, but only lowers it in the experimental group. How is this the case? Perhaps it does, but the authors didn't perform the comparison. It should KD the expression no matter what sample it is. I would be nice if the authors could at least provide the reviewers evidence demonstrating that it works in both groups, but of course the effect is more dramatic in the Tet3-GFAPcre samples.

We agree with the reviewer in that this is important to clarify.

Figure 4d represented the mRNA expression data in *Tet3-Gfap^{control}* and *Tet3-Gfap^{cre}* cultures after the interference with the shSNRPN using the 2-DDCt method. For this analysis we used *Tet3-Gfap^{control}* shSCRAMBLE as group of comparison. We have now represented the data using the 2-DcT method and performed the comparison in *Tet3-Gfap^{control}* before and after the interference with LV-shSNRPN. As shown in new Figure 4d the downregulation of *Snrpn* in *Tet3-Gfap^{control}* cells is also significant although, as the reviewer mentions, is more dramatic in *Tet3-Gfap^{cre}* cultures. Consistently, a significant increase in the number of secondary neurospheres, growth (new Figures 4f,g) and a decrease in the percentage of S100β+ cells (new Figures 4i, j) were observed in *Tet3-Gfap^{control}* cells after the interference of *Snrpn*.

10) The authors provide ample evidence in this manuscript that supports the idea that loss of Tet3 in SVZ NSCs disrupts self-renewal and skews that population towards increased levels of differentiated astrocytic cells, positive for S100β. Moreover, their data strongly suggest that this disruption appears to result from an apparent non-catalytic role for TET3 in regulating the expression of the imprinted gene *Snrpn*. As a consequence of increased *Snrpn*, BMP signaling is upregulated in SVC NSCs, which is sufficient to drive a larger than normal percentage of these cells to adopt an astrocytic fate.

Given that the authors examine this phenomenon on so many levels it would be extremely helpful to the reader to provide an illustration of their TET3-*Snrpn*-BMP model and how it regulates self-renewal and astrocytic fate in SVC NSCs (pretty much a visual abstract). Ideally, this would be found at the end of the results section, if space allows.

We have now included a graphical abstract summarizing the results described in the manuscript (new Figure S10).

Main changes made to Figures

References

Belenguer, G., Domingo-Muelas, A., Ferron, S. R., Morante-Redolat, J. M. & Farinas, I. Isolation, culture and analysis of adult subependymal neural stem cells. *Differentiation* **91**, 28-41, doi:10.1016/j.diff.2016.01.005 (2016).

Ferron, S. R. *et al.* A combined ex/in vivo assay to detect effects of exogenously added factors in neural stem cells. *Nat Protoc* **2**, 849-859 (2007).

Ferron, S. R. *et al.* Postnatal loss of Dlk1 imprinting in stem cells and niche astrocytes regulates neurogenesis. *Nature* **475**, 381-385, doi:10.1038/nature10229 (2011).

Luo, J., Daniels, S. B., Lenington, J. B., Notti, R. Q. & Conover, J. C. The aging neurogenic subventricular zone. *Aging cell* **5**, 139-152, doi:10.1111/j.1474-9726.2006.00197.x (2006).

Mirzadeh, Z., Doetsch, F., Sawamoto, K., Wichterle, H. & Alvarez-Buylla, A. The subventricular zone en-face: wholemount staining and ependymal flow. *Journal of visualized experiments : JoVE*, doi:10.3791/1938 (2010).

Raponi, E. *et al.* S100B expression defines a state in which GFAP-expressing cells lose their neural stem cell potential and acquire a more mature developmental stage. *Glia* **55**, 165-177, doi:10.1002/glia.20445 (2007).

Santos, F. *et al.* Active demethylation in mouse zygotes involves cytosine deamination and base excision repair. *Epigenetics Chromatin* **6**, 39, doi:10.1186/1756-8935-6-39 (2013).

REVIEWERS' COMMENTS:

Reviewer #1 (Remarks to the Author):

My concerns have been adequately addressed. I support publication.

Reviewer #2 (Remarks to the Author):

The authors have done a good job with the revised manuscript, my principal concerns appear to have been addressed adequately.

TET3 prevents terminal differentiation of adult NSCs by a non-catalytic action at Snrpn
NCOMMS-18-28439

Response to Reviewer Comments:

Each comment made the reviewers (highlighted in bold) is listed and followed by our response.

Reviewer #1:

My concerns have been adequately addressed. I support publication.

We thank the reviewer for his/her positive comments.

Reviewer #2:

The authors have done a good job with the revised manuscript, my principal concerns appear to have been addressed adequately.

We also thank this reviewer for his/her positive comments.